



# Modeling actinic flux and photolysis frequencies in dense biomass burning plumes

Jan-Lukas Tirpitz[1], Santo Fedele Colosimo[1], Nathaniel Brockway[2], Robert Spurr[3], Matt Christi[4], Samuel Hall[5], Kirk Ullmann[5], Johnathan Hair[6], Taylor Shingler[6], Rodney Weber[7], Jack Dibb[8], Richard Moore[6], Elizabeth Wiggins[6], Vijay Natraj[9], Nicolas Theys[10], and Jochen Stutz[1]

[1]Department of Atmospheric and Oceanic Sciences, UCLA, Los Angeles, CA, USA
[2]University of Alaska, Fairbanks, AK, USA
[3]RT Solutions Inc., Cambridge, MA, USA
[4]Fort Collins, CO, USA
[5]National Center for Atmospheric Research, Atmospheric Chemistry Observations and Modeling Laboratory, Boulder, Colorado, USA
[6]National Aeronautics and Space Administration, Langley Research Center, Hampton, Virginia, USA
[7]School of Earth and Atmospheric Sciences, Georgia Institute of Technology, Atlanta, GA, USA
[8]Earth Systems Research Center, University of New Hampshire, Durham, NH, USA
[9]Jet Propulsion Laboratory, California Institute of Technology, Pasadena, CA, USA
[10]Royal Belgian Institute for Space Aeronomy, Brussels, Belgium

**Correspondence:** Jan-Lukas Tirpitz (jltirpitz@atmos.ucla.edu)

**Abstract.** Biomass-burning (BB) affects air quality and climate by releasing large amounts of gaseous and particulate pollutants into the atmosphere. Photochemical processing during daylight transforms these emissions, influencing their overall environmental impact. Accurately quantifying the photochemical drivers, namely actinic flux and photolysis frequencies, is crucial to constrain this chemistry. However, the complex radiative transfer within BB plumes presents a significant challenge

for both direct observations and numerical models.

This study introduces an expanded version of the 1D VLIDORT-QS radiative transfer (RT) model, named VLIDORT for PhotoChemistry (VPC). VPC is designed for photochemical and remote sensing applications, particularly in BB plumes and other complex scenarios. To validate VPC and investigate photochemical conditions within BB plumes, the model was used to simulate spatial distributions of actinic fluxes and photolysis frequencies for the Shady wildfire (Idaho, US, 2019), based

on plume composition data from the NOAA/NASA FIREX-AQ (Fire Influence on Regional to Global Environments and Air Quality) campaign.

Comparison between modeling results and observations by the UCAR CAFS (Charged-coupled device Actinic Flux Spectroradiometer) yield a modeling accuracy of 10 - 20 %. Systematic biases between model and observations are within 2%, indicating that the uncertainties are most likely due to variability in the input data caused by the inhomogeneity of the plume as well

as 3D RT effects not captured in the model. Random uncertainties are largest in the ultra-violet (UV) spectral range, where they are dominated by uncertainties in the plume particle size distribution and brown carbon (BrC) absorptive properties.

The modeled actinic fluxes show a decrease from the plume top to bottom of the plume with a strong spectral dependence caused by BrC absorption, which darkens the plume towards shorter wavelengths. In the visible (Vis) spectral range, actinic





fluxes above the plume are enhanced by up to $60\,\%$. In contrast, in the UV, actinic fluxes above the plume are not affected or

even reduced by up to $10\,\%$. Strong reductions exceeding an order of magnitude in and below the plume occur for both spectral

ranges but are more pronounced in the UV.

# 1 Introduction

Biomass-burning (BB) constitutes a major source of particulate and gaseous atmospheric pollutants with significant impact on

air quality and climate (Crutzen and Andreae, 1990; Bond et al., 2013; Klimont et al., 2017). As a consequence of climate

change, natural BB events are expected to increase in frequency and intensity, thereby gaining even further significance in

the coming decades (Jaffe et al., 2020; McClure and Jaffe, 2018; O'Dell et al., 2019). Particles emitted from BB primarily

consist of black carbon (BC) and organic carbon (OC) (Reid et al., 2005; Levin et al., 2010) while emitted gases include

carbon monoxide, nitrogen oxides, a plethora of volatile organic compounds, and greenhouse gases such as $CO_2$, $CH_4$ and

$N_2O$ (Andreae, 2019). During daytime, both BB particles and gases undergo photochemical processing in the plume, leading

to the removal and transformation of emitted gases, formation of secondary pollutants, and changes in particle properties (Hand

et al., 2010; Forrister et al., 2015; Sumlin et al., 2017; Peng et al., 2021; Xu et al., 2021; Liu et al., 2021; Hennigan et al., 2012;

Cappa et al., 2020; Kiland et al., 2023). The main driver for these processes is photochemistry inside the plume, which is often

optically thick and highly inhomogeneous. Consequently, the accurate quantification of the photochemical drivers is crucial to

assess the impact of biomass-burning emissions on the environment.

Understanding and predicting the photochemistry in a given environment, requires knowledge of the speed at which the

involved reactions proceed. For a photochemical reaction

$$A + h\nu \rightarrow B + C, \tag{1}$$

initiated by a photon $h\nu$, the reaction rate

$$\frac{d[A]}{dt} = -J[A] \tag{2}$$

is determined by the photolysis frequency $J$ (expressed in Hz), which can be calculated according to

$$J = \int \sigma(\lambda)\Phi(\lambda)F(\lambda)d\lambda. \tag{3}$$

The absorption cross-section $\sigma(\lambda)$ of $A$ describes the probability of photons being absorbed, while the quantum yield $\Phi(\lambda)$

is the probability for absorbed photons to undergo the reaction. $\sigma(\lambda)$ and $\Phi(\lambda)$ are typically known from laboratory measure-

ments. The actinic flux $F(\lambda)$ describes the spherically integrated, directionally independent number of photons at wavelength

$\lambda$. $F(\lambda)$ is controlled by environmental factors, such as altitude, solar position, cloudiness, aerosols and Earth surface proper-





ties. All of these conditions are temporally and spatially variable. Consequently, $F(\lambda)$ is highly variable and its determination can be challenging.

Direct measurements of $F(\lambda)$ can be performed with high accuracy using instruments with direction-insensitive recep-
tion optics, and spectrometers that cover the required spectral and dynamic range (e.g. Junkermann et al., 1989; Kraus and Hofzumahaus, 1998; Shetter and Müller, 1999; Bais et al., 2003). However, such measurements can only be performed in-situ and are thus limited in spatio-temporal coverage. Another, often complementary, approach is to simulate actinic fluxes using radiative transport (RT) models. With such models, continuous spatial distributions of actinic fluxes can be derived. RT models need to be constrained with detailed information on the atmospheric conditions, in particular on the spatial distribution and
properties of clouds and aerosols. In chemical transport models the inputs for the built-in RT models are driven by the chemical transport simulation itself. For the interpretation of field experiments and observations, the compilation of adequate RT model input data is more challenging.

Over the past 50 years, both direct measurements of actinic fluxes and RT simulations have been applied to obtain a complete picture of the radiative conditions for clear sky conditions (e.g. Turco, 1975; Meier et al., 1997; Shetter et al., 2002; Volz-
Thomas et al., 1996; Hofzumahaus et al., 2002; Wagner et al., 2011) and above, below, and inside clouds (e.g. Thompson, 1984; Madronich, 1987; Junkermann, 1994; Kelley et al., 1995; Lefer et al., 2003; Tie et al., 2003; Liu et al., 2006; Neu et al., 2007; Ryu et al., 2017). In contrast, for BB plumes, corresponding studies rely on observations only, whereas the modeling of actinic fluxes and photolysis frequencies and their spatial distribution has received less attention. For instance, many current experimental BB plume chemistry studies are based on directly measured actinic fluxes (Decker et al., 2021; Liao et al.,
2021) or irradiances (e.g. Lindsay et al., 2022), or on the deduction of actinic fluxes from concentration measurements of gases involved in well-known proxy reactions (Peng et al., 2021). Such measurements are however limited in spatio-temporal coverage as they can only provide information at a single location and time. Continuous distributions of actinic flux and photolysis frequencies in BB plumes with the accuracy and spatial resolution needed for atmospheric chemistry studies can only be inferred using RT models (e.g Decker et al., 2021; Palm et al., 2021). Trentmann et al. (2003a; 2003b) performed such
RT modeling, but only for a synthetic BB plume, created with a three-dimensional chemical transport model. To our knowledge, similar studies for real plume scenarios do not currently exist. This is not surprising, as modeling is typically challenged by a lack of information to constrain the RT model, arising from the complex and variable nature of such plumes, in particular when they are young (hours old) and optically dense. The RT in BB plumes is dominated by the aerosol, for which the spatial distributions and optical properties are often not well-known. A peculiarity in BB plumes is the optically active fraction of OC,
typically referred to as "brown carbon" (BrC), which becomes strongly absorbing towards shorter wavelengths (Laskin et al., 2015). It dominates the RT in the ultra-violet (UV) spectral range, where photochemistry is most responsive. Its concentrations and optical properties are difficult to assess, as they depend on the burned fuel, burning phase and smoke age (Forrister et al., 2015; Laskin et al., 2015; Sumlin et al., 2018; Zeng et al., 2022; Shetty et al., 2023).

In addition, challenges arise from the modeling side. Simulation of horizontal radiative transport effects, such as side il-
lumination and shadowing require three-dimensional RT models (e.g. Mayer, 2009; Deutschmann et al., 2011), which are computationally expensive. Most models, including the model used in this study, therefore assume a horizontally homoge-



neous atmosphere; they solve the radiative transfer equation in one dimension and are thus referred to as "1D RT models". This assumption strongly increases efficiency but can introduce errors due to the inability to simulate RT in the horizontal dimension. Investigations of these effects have been performed based on synthetic data (Trentmann, 2003) showing the significance

of shadowing. 1D RT models, and to a certain extent 3D-models, often suffer from the incomplete knowledge and simplified description of the input data, particularly in complex environments where atmospheric inhomogeneities can introduce uncertainties in the model initialization. These inhomogeneity effects and challenges with horizontal RT in real BB plume scenarios have not been well studied. There is thus a need to better understand how well 1D RT models can describe actinic fluxes and photolysis frequencies in these plumes.

The accuracy of modeled actinic fluxes and photolysis frequencies can only be ensured by comparison with established models or with measurements. While such comparisons have been conducted extensively for well-defined atmospheric scenarios (e.g. Barnard et al., 2004; Kelley et al., 1995; Castro et al., 1997; Volz-Thomas et al., 1996; Dickerson et al., 1997; Kazantzidis et al., 2001; Vuilleumier et al., 2001; Balis et al., 2002; Shetter et al., 2003; Hofzumahaus et al., 2004), they have not been performed for the complex radiative conditions encountered in optically dense BB plumes.

In the present study we derive continuous spatial distributions of actinic flux and photolysis frequencies in a real and optically dense BB plume, by means of RT simulations. The key objectives of the study are:

1. Introduce and validate a recently developed radiative transfer model (VPC) for photochemical and remote sensing applications in BB plumes.

2. Assess the viability and limitations of actinic flux and photolysis modeling in BB plumes, using 1D RT models and
state-of-the-art plume composition measurements.

3. Provide insights into spatial actinic flux and photolysis frequency distributions in BB plumes.

The VPC model (Section 2) is an extended version of the VLIDORT-QS model (Spurr, 2006), a variant of the widely-used VLIDORT RTM. VPC can calculate actinic fluxes based on a range of input parameters. We constrain VPC using recent airborne BB plume composition measurements performed during the FIREX-AQ campaign in 2019 (Section 3), to simulate 2D
distributions of actinic flux and photolysis frequencies over 20 plume cross-sections (Section 4). We compare these modeling results to FIREX-AQ actinic flux and photolysis frequency measurements to validate the VPC results (Section 5) and to quantify systematic and random uncertainties. We investigate spectral and environmental dependencies, and perform model sensitivity studies to identify critical factors that limit the modeling accuracy. In addition, we investigate selected model results to provide deeper insight into the spatial and spectral distribution of actinic fluxes and photolysis frequencies in BB plumes (Section 6).
Potential implications and future studies in BB plume radiative transport, chemical modeling and remote sensing applications are discussed at the end of this paper (Section 6).



## 2 Model description

VLIDORT for photochemistry (VPC), is built around the Quasi-Spherical Vector Linearized Discrete Ordinate Radiative Transfer (VLIDORT-QS) code (Section 2.1). The VLIDORT family of RT codes are widely used in the trace gas remote sensing community (Spurr, 2006; Spurr and Christi, 2019). These codes have been tested for many different atmospheric conditions and are thus well-suited for the study and interpretation of field observations. While there are other well-established and efficient RT models for the calculations of photolysis frequencies, such as the Troposheric UV and Visible (TUV) radiation model fortran code (Madronich and Flocke, 1999)) or Fast-J (Wild et al., 2000), VPC particularly facilitates the interpretation of field observations of BB plumes. To allow the study of dense BB plumes, VPC has to meet the following requirements: (1) a suitable and flexible representation of BB plume particles in the model, (2) efficient retrieval and sensitivity analyses common in remote sensing studies, (3) accurate RT results for a wide variety of geometries, including high solar zenith angles (SZA) and limb viewing directions, (4) simulation of the light's polarisation state, allowing for retrieval of aerosol information from polarimetric remote sensing data (which will be explored in future studies) and (5) the simultaneous calculation of radiances and actinic fluxes, useful for applications where actinic fluxes are deduced from radiance observations, or where photochemistry and remote sensing are closely linked (e.g. for plume composition measurements from satellite).

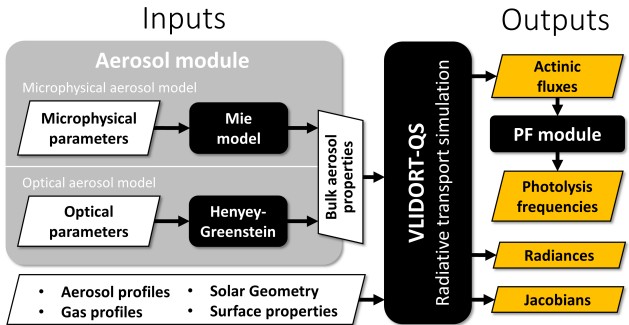

**Figure 1.** Schematic of the VPC model.

An overview of the model is depicted in Fig. 1. On the input side, VPC features the Aerosol module, which can account for multiple types of aerosol in a flexible way (Section 2.2.2). On the output side, a module has been added to convert the RT-simulated actinic flux spectra to photolysis frequencies for various chemical reactions (Section 2.4). VLIDORT-QS, the photolysis frequency module and parts of the aerosol module are implemented in Fortran, but have been embedded in object-oriented Python wrappers, to obtain a structured and modular user interface, including useful functions for data pre- and post-processing as well as visualization. The package supports not only combined use but also stand-alone individual use of the Aerosol, VLIDORT-QS and Photolysis frequency modules. The current model default configuration is optimized for tropospheric applications in the wavelength range between 295 and 650 nm. Extension to stratospheric applications requires adding additional absorbers (e.g. $O_2$ absorption in the deep UV).





## 2.1 VLIDORT-QS radiative transport model

The core of VPC is the VLIDORT-QS radiative transfer code, which is designed for a spherically curved atmosphere. Single scattering (or first-order radiative transfer) is treated exactly for a spherically curved medium, while multiple scattering radiation fields along a given line-of-sight are calculated using a modified version of the standard VLIDORT model (Spurr, 2006) working in plane-parallel scattering mode. A detailed description of the VLIDORT-QS model, including validations against 3D Monte-Carlo RT models, can be found in Spurr et al. (2022).

Since the original work on VLIDORT-QS was reported, we have added a number of additional features to the model. Most important for the present study, is the ability to generate directionally integrated fluxes (actinic fluxes and irradiances) at all segment boundaries. The model also features a complete linearization scheme, i.e. in addition to the radiation field itself VLIDORT-QS will generate a complete range of analytically-derived radiance or Stokes-vector Jacobians (weighting or sensitivity functions) with respect to any atmospheric parameter, including aerosol loading and optical properties in BB plumes. This capability makes VLIDORT-QS suitable for retrieval and sensitivity analyses common in remote sensing studies, and adds this ability to observations of actinic fluxes. Finally a number of performance enhancements have been made in order to speed up the calculations; in particular the code has been made thread-safe for use in parallel-computing environments such as OpenMP.

## 2.2 Model inputs

Table 1 provides an overview on the most relevant model input parameters, which are discussed in more detail in the following subsections.

### 2.2.1 Vertical grid

In VPC the atmosphere is represented by a finite number of stacked optically homogeneous layers. Common layer numbers are on the order of 100 over an altitude range from zero to about 70 km ASL. Layer thicknesses typically increase with altitude from about 100 m at the surface to several km at the top of the atmosphere (e.g. Madronich and Flocke, 1999). Per default, heights $h_l$ for each layer boundary $l = \{0, 1, ..., l_{\max}\}$ are defined via $h_l = c_1(c_2^l - 1)$, resulting in an exponential grid with increasing layer thickness, but VPC also allows for user defined arbitrary vertical grids.

### 2.2.2 Aerosol module

In the RT simulation performed with VLIDORT-QS, particles are described by a single set of bulk aerosol optical properties, namely the extinction coefficient, single scattering albedo (SSA) and scattering phase matrix. The purpose of the aerosol module is to pre-calculate these quantities for mixtures of aerosol types with different properties. The module allows to include an arbitrary number of aerosol types in the RT simulation. As described in detail in the following paragraphs, for each aerosol type the user chooses between two available aerosol models, provide aerosol properties as required by the respective model, and defines the aerosol vertical distribution.





**Table 1.** Overview of VPC input parameters. Index $l$ indicates parameters defined individually for each atmospheric model layer. $\lambda$ indicates dependence on wavelength (see Section 2.5 for further details). Aerosol parameters are defined for each aerosol type $a = \{1, 2, 3, ...\}$. Parameters with an index $m = \{1, 2, 3\}$ need to be specified for each mode of the trimodal PSD.

| Subgroup | Symbol | Description | See also |
|---|---|---|---|
| Microphysical | $r_{m,a}$ | PSD mode median radii | |
| aerosol | $\sigma_{m,a}$ | PSD modal widths | |
| module | $f_{m,a}$ | Fraction of particles residing in first two PSD modes ($m = \{1, 2\}$). Third mode fraction | |
| | | $f_3 = 1 - f_1 - f_2$ results from PSD normalisation. | |
| | $n_a(\lambda)$ | Particle real refractive index | |
| | $\kappa_a(\lambda)$ | Particle imaginary refractive index | |
| Optical | $\omega_a(\lambda)$ | Aerosol single scattering albedo (SSA) | |
| aerosol | $g_a(\lambda)$ | Asymmetry parameter for Henyey-Greenstein scattering phase function | |
| module | $\alpha_{i,a}^{(E)}$ | wavelength-dependence coefficients for aerosol extinction coefficient $E$ | Eq. 4 |
| Profile | $h_l$ | Model vertical grid (altitudes above ground of atmospheric layer boundaries) | |
| information | $p_l$ | Pressure | |
| | $T_l$ | Temperature | |
| | $E_{l,a}(\lambda_0)$ | Aerosol extinction coefficients at reference wavelength $\lambda_0$ | |
| | $c_l^{(s)}$ | Number concentration of variable number of trace gases $s = \{O_3, NO_2, SO_2, ...\}$ | |
| Surface | $A(\lambda)$ | Lambertian surface albedo | |
| Other | $\lambda_w$ | List of simulated wavelengths | |
| | $\Delta\lambda_{FWHM}$ | Spectral averaging | Section 2.5 |
| | | Solar zenith angle | |
| | | Viewing geometry* (viewing elevation, relative azimuth, observer altitude) | |

\* Of relevance for radiance calculations only.

The two available aerosol models are in the following referred to as *microphysical* and *optical* aerosol model respectively (Fig. 1). The *microphysical aerosol model* describes aerosol in terms of a particle size distribution (PSD) and complex particle refractive index (real part $n$ and imaginary part $\kappa$). These are fed to an integrated Mie aerosol model (Spurr et al., 2012) to derive exact bulk optical properties, including the exact scattering phase matrices, assuming spherical particles. The PSD is 170 tri-modal, with up to three log-normal distributions. The (normalized) number of particles residing in each mode is described by the modal fractions $f_m$, with $f_1 + f_2 + f_3 = 1$. In the future, the microphysical model will be expanded to account for coated or non-spherical particles (see e.g. Kattawar and Hood, 1976; Borghese et al., 1979; Mackowski and Mishchenko, 1996; Muinonen et al., 1996).



For some applications, e.g. remote sensing retrievals with limited information in the measurements, it is useful to stay in
the optical domain and skip the additional layer of complexity added by a microphysical aerosol representation. This is the
purpose of the *optical aerosol model*, wherein aerosol is described by the wavelength-dependence coefficients $\alpha_{i,a}^{(E)}$ (Eq. 4)
and the SSA $\omega_a(\lambda)$. The scattering phase function is defined via the scattering asymmetry parameter (AP) $g_a(\lambda)$, assuming a
Henyey-Greenstein formalism (Henyey and Greenstein, 1941).

The properties of each aerosol type are constant with altitude. Altitude dependencies are realized via superposition of multiple aerosol types with different vertical profiles. To simplify the interpretation of field observations in VPC, aerosol amounts
are expressed in terms of the aerosol extinction coefficient $E_{l,a}(\lambda_0)$ (extinction per km). Extinction coefficients are often provided by remote sensing observation. They are fed to the model for a specific reference wavelength $\lambda_0$, for each model layer $l$
and each aerosol type $a$.

### 2.2.3 Trace gas absorption

For the radiative transport simulation, VPC includes literature cross-sections for $O_3$, (Brion et al., 1998) $O_4$, (Thalman and
Volkamer, 2013), $H_2O$ (Rothman et al., 2010; Lampel et al., 2015) and $NO_2$ (Vandaele et al., 1998). Users can provide vertical
concentration profiles for each of these gases to account for their absorption. VPC also facilitates the implementation of
additional gases.

### 2.2.4 Atmospheric boundaries

For the extraterrestrial solar irradiance at the top of the atmosphere (TOA), VPC users can choose to either use the spectrum by
Chance and Kurucz (2010) or by Coddington et al. (2023). Surface properties in VLIDORT-QS can be defined via a Lambertian
equivalent reflectance (LER) or providing the full bidirectional reflectance distribution function (BRDF). In the current VPC
framework only the wavelength-dependent LER is implemented. If required, BRDF functionality might be added in the future.

### 2.3 Wavelength-dependent parameters

Some model input parameters depend on wavelength, such as refractive indices, albedos, extinction coefficient and asymmetry parameter. In this section we refer to them by $x(\lambda)$. For each wavelength-dependent parameter, the model offers two
approaches:

1. Providing independent values for a set of discrete wavelengths.

2. Providing a single value $x_0$ for a reference wavelength $\lambda_0$ and coefficients $\alpha_i$ for an Angstrom wavelength-dependence
   of arbitrary order $i_{\max}$.

In the second case, parameter values for each wavelength are internally calculated assuming:

$$\ln\left(\frac{x}{x_0}\right) = \sum_{i=1}^{i_{\max}} \alpha_i \left(\ln\frac{\lambda_0}{\lambda}\right)^i \tag{4}$$



This parameterisation represents an extended version of the classic Ångstrom wavelength-dependence (Ångström, 1929). In fact, for $i_{max} = 1$, Eq. 4 can be rearranged to obtain the well-known dependence:

$$\frac{x}{x_0} = \left(\frac{\lambda}{\lambda_0}\right)^{-\alpha_1}, \tag{5}$$

Increasing $i_{max}$ to two, adds a quadratic term to Eq. 4, effectively accounting for a log-linear wavelength-dependence of the Ångstrom parameter, similar to parameterisations proposed by King and Byrne (1976); Kaufman (1993) or Schuster et al. (2006). Increasing $i_{max}$ adds higher order wavelength-dependencies. The parameterisation was chosen because of its general applicability. We found that Eq. 4 can successfully capture wavelength-dependencies not only of extinction but also other parameters, such as SSA, AP and aerosol refractive indexes measured during FIREX-AQ (see Supplement S1). In contrast to the conventional Ångstrom dependence, Eq. 4 can also describe the strong non-log-linear wavelength-dependence of BrC absorption in BB plumes.

As decribed in Section 2.2.2, aerosol vertical profiles are provided in terms of extinction coefficient $E(\lambda_0)$ at a reference wavelength $\lambda_0$. The wavelength-dependence of $E(\lambda)$ is either derived from internal Mie model calculations, when using the microphysical aerosol model, or according to Eq. 4, when using the optical aerosol model. In the case of the SSA $\omega$, Eq. 4 was found to better describe (i.e. with lower $i_{max}$ values) the single scattering co-albedo $1 - \omega$ instead of $\omega$ itself. Thus, in the optical aerosol model Eq. 4 is applied to $1 - \omega$.

### 2.4 Model outputs

The main outputs of a RT simulation with VLIDORT-QS are actinic flux spectra $F_l(\lambda)$ at each layer boundary of the model's vertical grid. Separate spectra are provided for three contributions to $F_l(\lambda)$:

1. Direct solar beam, attenuated by atmospheric extinction

2. Downwelling diffuse radiation, describing the amount of scattered photons incident from the hemisphere above the observer

3. Upwelling diffuse radiation, describing the amount of scattered photons incident from the hemisphere below the observer

This separation is useful as it allows for deeper analysis of the results, also considering that real measurements are typically performed separately for upwelling (diffuse only) and downwelling (direct + diffuse) actinic fluxes.

In addition, radiance spectra for a prescribed viewing geometry are calculated in the same simulation process (see Table 1). VLIDORT-QS and the Mie model feature an analytical linearisation scheme (see e.g. Spurr et al., 2022), which can provide Jacobians (sensitivities) of the outputs with respect to any input parameter with high computational efficiency.

For the conversion of actinic flux spectra to photolysis frequencies (performed by the "PF-module", see Fig. 1), we isolated and adapted a corresponding module from TUV (version 5.4) (Madronich and Flocke, 1999). It reads and prepares all required cross-sections and quantum yields to calculate photolysis frequencies for 113 photochemical reactions based on Eq. 3. Using this established code as basis ensures consistency with the atmospheric chemistry community. We made a few adaptations: we



added missing data on the $N_2O_5$ absorption cross section for the 340 to 410 nm range, and we adapted $ClNO_2$ cross section
and quantum yield based on the 2015 JPL recommendations (Burkholder et al., 2019). Furthermore, we extended the module
by including 14 additional reactions for halogen chemistry.

The TUV module was also used to process the measured actinic flux spectra from FIREX-AQ (see Section 3.3). Note that
for the present study, only 48 photolysis reactions relevant to the spectral range of the measurements were considered. An
overview of all of the reactions available is provided in Supplement S5.

## 2.5 Spectral averaging and interpolation

RT simulations can be performed for user-defined sets of wavelengths. However, using very small wavelength intervals is
inefficient, while the accuracy of the simulation suffers if too few wavelength are chosen. A number of steps have been taken
to make the model more efficient. The resolution of the originally highly-resolved ($\Delta\lambda \approx 0.01$ nm) literature spectra used by
the model (Section 2.2.3 and 2.2.4) can be reduced prior to simulation by Gaussian smoothing. Although this is not physically
correct (ideally, smoothing is applied to the line-by-line modeled spectrum after simulation), this option is useful for efficient
calculation of outputs averaged over broader wavelength intervals at the cost of relatively small introduced errors. We found
$< 1\%$ ($< 3\%$) error in photolysis frequencies for a smoothing kernel of 1 nm (2 nm) FWHM and typical atmospheric scenarios.

To improve efficiency further, a novel spectral interpolation approach was developed; this allows reproduction of high res-
olution ($\Delta\lambda \approx 1$ nm) actinic flux spectra from simulations at a few suitable wavelengths. Several models use approaches for
efficient photolysis rate calculation by reducing the number of full radiative transport calculations to a few wavelengths (<10)
(e.g. Landgraf and Crutzen, 1998; Williams et al., 2006; Wild et al., 2000; Madronich and Flocke, 1999). They are however
limited in accuracy ($\approx 10\%$ in photolysis frequencies in the UV) and optimized for applications in clean atmospheres. Our ap-
proach uses more wavelengths (10 to 30, depending on SZA and atmospheric conditions) and is thus slower but more accurate
($< 2\%$ error in photolysis frequencies) and it provides high resolution spectra as an intermediate product. Furthermore, it also
works for the particular conditions encountered in this study, including the presence of the strongly wavelength-dependent BrC
absorption and high SZAs.

Our approach is inspired by the passive Differential Optical Absorption Spectroscopy (DOAS) measurement technique (Platt
and Stutz, 2008). The passive DOAS approach measures radiances of scattered skylight, but the basic concepts described in
the following hold for both radiances and actinic fluxes. Most of the structures observed in an actinic flux spectrum $F(\lambda)$ are
solar Fraunhofer lines already present in the extraterrestrial solar irradiance spectrum $I_0$. The actual atmospheric signal can be
isolated and conveniently described by the actinic flux optical depth (AFOD)

$$\tau(\lambda) = -\ln\left(\frac{F(\lambda)}{I_0(\lambda)}\right). \tag{6}$$

We make use of the fact that only trace gas absorption introduces significant narrow-band features in $\tau(\lambda)$, whereas scattering
and aerosol absorption impose a spectrally smooth signal. Therefore, just as for passive DOAS radiance OD spectra, AFOD





spectra can be approximated by:

$$\tau(\lambda) = \sum_{i=0}^{n_p} P_i \lambda^i + \sum_s S_s(\lambda) \cdot \sigma_s(\lambda, T_s). \tag{7}$$

Here, the first term represents a polynomial that accounts for the spectrally smooth scattering and aerosol absorption signals. The second term accounts for optical depth contributions of each trace gas species $s$ with absorptions strong enough to significantly influence actinic fluxes. In this study and in most other cases, this is only ozone. $\sigma^{(s)}(\lambda, T_s)$ are the trace gas absorption

cross-sections from literature, which can depend on the gas temperature $T_s$. $S^{(s)}(\lambda)$ is the observed slant column density (SCD), which is the gas' concentration integrated along the effective light path. The term "effective" is used here to indicate that contributing photons travel along an infinite number of light paths, each with a distinct probability. These probabilities change with wavelength, as do the effective lightpath lengths and the SCD. Following Pukīte et al. (2010), we parameterise the SCD wavelength-dependence by

$$S(\lambda) = \sum_{j=0}^{n_\lambda} S_{\lambda,j} \lambda^j + \sum_{k=1}^{n_\sigma} S_{\sigma,k} \sigma(\lambda), \tag{8}$$

where we omitted the index $s$ for readability. The first sum is a polynomial, with coefficients $S_\lambda$. It describes the SCD including a spectrally smooth wavelength-dependence that arises from changes in RT due to scattering and aerosol absorption. $S_\sigma$ accounts for changes in effective light path due to strong narrow-band gas absorption, which favours short light path lengths through the absorbing gas.

If $O_3$ is the only absorbing gas of relevance, Eq. 7 and Eq. 8 describe any actinic flux spectrum based on $n_p + n_\lambda + n_\sigma + 1$ free parameters, including $T_s$. These parameters can be inferred by fitting Eq. 7 to simulated AFODs at a few suitable wavelengths. From the retrieved parameter values, the full high resolution spectrum can be reconstructed.

The optimization of the interpolation settings applied for this study, as well as an accuracy assessment, are described in detail in Supplement S2. In short, we use simulations at 23 wavelengths to reproduce high resolution actinic flux spectra

($\Delta\lambda = 2\,\text{nm}$ from 298 to 640 nm). Typical differences to the exact line-by-line calculated spectra were assessed for a wide range of conditions, yielding a root-mean-square deviation (RMSD) of $0.3\,\%$. Maximum differences are observed in the UV and for SZAs $> 80°$ but never exceed $7\,\%$. Resulting errors in photolysis frequencies are even smaller, with a RMSD of $0.2\,\%$ and a maximum of $1.8\,\%$. We therefore conclude that errors introduced by the interpolation are negligible compared to other sources of uncertainty encountered in the present study.

In the future, more investigations, similar to those in Supplement S2, can be performed to further enhance efficiency and make the interpolation generally applicable for other spectral ranges, spectral resolution and conditions.

## 3 Measurement data

The present study is based on data from the 2019 NOAA/NASA FIREX-AQ measurement campaign (Warneke et al., 2023). In the course of this 6-week airborne campaign, more than 90 BB plumes were sampled in-situ as well as remotely from the





NASA DC-8 research aircraft. For our purpose, the FIREX-AQ observations offer a unique opportunity, as they provide a view of plume composition and radiative conditions of unprecedented comprehensiveness and accuracy. An overview of all of the instruments aboard the aircraft and the raw data is available in the NASA FIREX-AQ data archive (Aknan and Chen, 2023). This section provides basic information on the flight selected for our analysis and instrumentation relevant for our study.

### 3.1 Shady Fire Overview

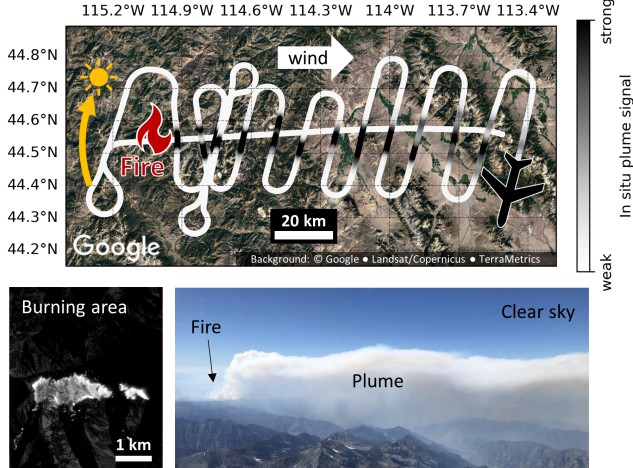

**Figure 2.** Upper panel shows Shady fire geolocation and a part of the flight track (see red box in Fig. 3), including an overflight along the plume axis and following transects. Approximate location and movement of the Sun during the flight are indicated on the left. Lower left shows a Nadir infrared image of the burning area (Aknan and Chen, 2023), recorded from the aircraft at the end of the second overflight.

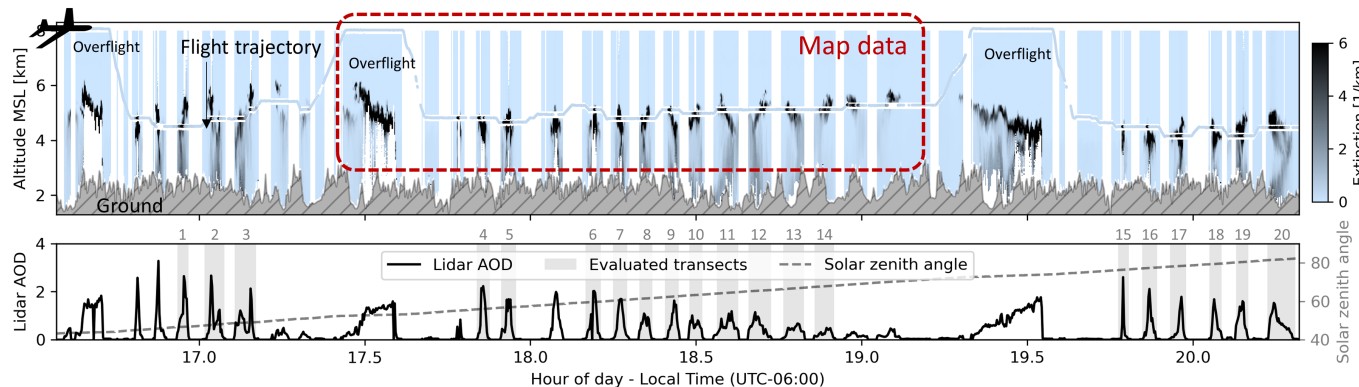

**Figure 3.** The Shady fire sampling flight profile. Top panel shows aerosol extinction profiles from the lidar. The flight track is embedded and color coded based on in-situ aerosol extinction measurements. Lower panel shows the lidar aerosol optical depth (AOD) at 532 nm and solar elevation angle. Grey shaded areas indicate the evaluated transects. The red box indicates the data which is visualized on the map in Fig. 2.



Our case study focuses on measurements from the "Shady Fire" on July 25, 2019 in Idaho because the burned area over the sampling period was small ($\approx 2\,\mathrm{km}^2$) and the burned fuel homogeneous. The plume was large with an extent of about $10 \times 1 \times 100\,\mathrm{km}$ (W×H×L), ensuring good spatial sampling despite the high aircraft speed ($\approx 150\,\mathrm{m/s}$) and justifying the 1D model assumption of a horizontal homogeneous atmosphere. Observed solar azimuth angles (SAAs) were between $250°$ and $290°$, and the Sun is therefore almost aligned with the plume axis ($270°$ azimuthal orientation) over the entire flight (Fig. 2),

which is a favourable configuration for avoiding horizontal radiative transport effects (Section 6). Compared to other fires, the data coverage is very high, allowing for a detailed analysis and validation of the VPC model. The multiple-hour long flight included three plume overflights as well as 20 plume transects with most instruments collecting data.

    Fig. 2 and Fig. 3 provide a general overview of the conditions, fire location and flight path. Plume overflights were performed against the wind along the plume axis and were followed by plume transects with sequentially increasing distance to the fire.

Actinic flux and photolysis simulations were performed for each of the transects labeled in Fig.3. Some transects were excluded due to gaps in crucial measurement data. Data from overflights were used only to infer the aerosol lidar ratio (see Section 4.2).

## 3.2   Measurements of plume properties

Measurements of plume geometry, composition and aerosol properties were used to constrain the VPC model.

    A combined High Spectral Resolution Lidar (HSRL) and an ozone DIfferential Absorption Lidar called DIAL-HSRL (Hair

et al., 2008; Browell, 1989) measured aerosol backscatter coefficient at $532\,\mathrm{nm}$ at a temporal resolution of $10\,\mathrm{s}$ (translating into a horizontal spatial resolution of about $1.5\,\mathrm{km}$) and vertical resolution of $30\,\mathrm{m}$ during FIREX-AQ. Measurements were performed simultaneously in the upward and downward directions. Profiles range from near the surface to about $8\,\mathrm{km}$ altitude, unless parts of the atmosphere are shielded by optically opaque plume layers. Although the backscatter coefficient is determined from a ratio of two channels with the same viewing geometry, for this system, the profiles contain a vertical gap of about

$250\,\mathrm{m}$ (see also Figure 3) extending above and below the aircraft due to changes in the geometrical overlap of the telescope and spatially dependent gains of the two photo-detectors that are used for the backscatter coefficient measurements (Wandinger and Ansmann, 2002; Simeonov et al., 1999). The lidar independently provides aerosol extinction profiles, however, at a reduced temporal and vertical resolution (60s and 100m) and with an increased geometrical overlap (> 2km around the aircraft Hair et al., 2008).

A TSI model 3340 laser aerosol spectrometer measured aerosol size distributions at $1\,\mathrm{s}$ temporal resolution, at aircraft cabin temperature and dry humidity. The data are reported for standard temperature and pressure and dry humidity The measured size range covered particle radii between $50\,\mathrm{nm}$ and $2.5\,\mathrm{microns}$. TSI Nephelometers, a Radiance Research Particle Soot Absorption Photometer (PSAP) and the NOAA Aerosol Optical Properties Suite (AOP, Langridge et al., 2011; Lack et al., 2012) were used to determine the in-situ aerosol absorption, scattering and extinction coefficients at a temporal resolution of $1\,\mathrm{s}$ and

various wavelengths. Measurements at 405, 532, and 664 nm wavelength were performed under dry and humidified conditions to determine the hygroscopicity factor, which was then used to scale the measurements to ambient humidity. Measurements at 450, 550, 532 and 700 nm were performed at dry conditions only.



BrC particle absorption between 300 and 700 nm was investigated with a technique in the following referred to as SAEB (spectral analysis of extracted BrC chromophores). For each plume transect particles were gathered on separate filters, which were later extracted with water and afterwards methanol (Liu et al., 2014, 2015; Zeng et al., 2020, 2022). While most organic material dissolves in either of the solvents, unsoluble black carbon remainders can be removed from the solution using pore filters. Spectral analysis of the remaining BrC chromophore solution in a Liquid Waveguide Capilary Cell (LWCC) then provides BrC absorption spectra, which we used to infer the wavelength-dependence of the BrC imaginary refractive index (see Section 4.2).

Carbon monoxide (CO) in-situ measurements from the DACOM instrument (Sachse et al., 1987) were used as plume indicator and for interpolation of gaps in aerosol extinction data (Section 4.2). $O_3$ was measured with a nitric oxide chemiluminescence monitor. The DC-8 aircraft's meteorology and navigation systems provided temperature, pressure, humidity, wind speed, wind direction, geolocation, altitude ASL, ground speed, and radar measured altitude above ground at 1 s temporal resolution.

### 3.3 Measurements of actinic fluxes and photolysis frequencies

For the validation of our modeling results, we use actinic flux spectra measured on the aircraft with Charged-coupled device Actinic Flux Spectroradiometers (CAFS). CAFS instruments measure in-situ down- and up-welling radiation and combine them to provide $4\pi$ steradian actinic flux density spectra from 298 to 640 nm. (Shetter and Müller, 1999; Hall et al., 2018). The sampling resolution is $\approx 0.8$ nm with a FWHM of 1.7 nm at 297 nm. The absolute spectral sensitivity of the instruments was determined in the laboratory with 1000 W NIST-traceable tungsten-halogen lamps with a wavelength-dependent uncertainty of 3 to 5 %. During deployments, spectral sensitivity drift was assessed with secondary calibration lamps while wavelength assignment was monitored with Hg line sources and comparisons to spectral features in the extraterrestrial flux. The optical collectors were characterized for angular and azimuthal response and the effective planar receptor distance. For FIREX-AQ, upgraded electronics and cooling improved the signal to noise allowing for 1 Hz acquisition. From the measured actinic flux, photolysis frequencies are calculated for 48 atmospheric trace gases (listed in Supplement S5), using the same module from the TUV model as VPC (Section 2.4).

In the following analysis, CAFS actinic fluxes are shown in conjunction with a confidence interval, consisting of two uncertainty contributions: a limit of detection of $6 \times 10^{10}$ photons s$^{-1}$ cm$^2$ nm$^{-1}$, estimated from noise analysis under low light conditions, and a span error of 5 %, arising from uncertainties in the instrument's calibration.

### 4 Model setup

To initialize our model, the measurements introduced in Section 3.2 were preprocessed and combined with literature and satellite data as described in the following subsections. An overview of the input data is provided in Table 2. We use a vertical grid of 87 layers extending from ground to 62 km above ground level (AGL). The layer thickness was set to 100 m between zero and 6 km (approximately covering the altitude range of the flight), 250 m between 6 and 9 km, 3000 m between 9 and 42 km, and 5000 m between 42 and 62 km altitude AGL. The spectrum by Chance and Kurucz (2010) was used for the



extraterrestrial solar irradiance. All literature spectra were spectrally smoothed prior to simulation ($\Delta\lambda = 2\,\text{nm}$) as described in Section 2.5, which was found to approximately match the resolution of the CAFS instrument. Full simulations were performed on an irregular wavelength grid with 23 nodes (see Supplement S2), with denser sampling towards UV wavelengths. The grid was optimized for the interpolation approach described in Section 2.5 and Supplement S2. The final actinic flux spectra were calculated from these simulations using the aforementioned interpolation approach with a spectral sampling interval of $0.2\,\text{nm}$.

For the VLIDORT scattering RT solver, the number of discrete-ordinate streams in the polar half-hemisphere was set to 8. In addition, the "delta-M scaling" ansatz was used to deal with sharply-peaked forward scattering characteristic of aerosols.

Spatial distributions (2D plume cross-sections) of actinic flux and photolysis frequencies are modeled for each of the 20 transects highlighted in the lower panel of Fig. 3. The horizontal resolution of the modeled distributions is determined by the speed of the aircraft and the temporal resolution at which input data is available. As indicated in Table 2, different model input 375 parameters are updated at different intervals:

1. 10 seconds: the temporal resolution of the model simulations is ultimately limited by the resolution of the lidar backscatter profile measurements ($10\,\text{s}$). All model input data available at shorter time spans are therefore averaged to at least $10\,\text{s}$ intervals prior to simulation, which corresponds to an approximate horizontal resolution of $1.5\,\text{km}$.

2. Per transect: in general BB-plumes exhibit high temporal and spatial variability, but many properties show little (or 380 at least undetectable) variation over individual transects (e.g. PSD, particle refractive indices and optical black carbon fraction). Some observations are thus averaged per transect to reduce measurement noise.

3. Fixed: for some parameters constant values are used for the entire flight, e.g. for those taken from literature or in the case of scarce data coverage.

### 4.1 Background aerosol

Two aerosol types are used to represent tropospheric and stratospheric background aerosol, both types based on the microphysical model of the VPC aerosol module (Section 2.2.2). For stratospheric background aerosol, extinction profiles were taken from the SAGE II database (SAGE Science Team, 2012). The size distribution was adapted from Wrana et al. (2021). For the particle refractive index we assume a mixture of $75\,\%$ $H_2SO_4$ and $25\,\%$ of water (Levoni et al., 1997), yielding a value of $1.43 + i \cdot 10^{-8}$, based on Palmer and Williams (1975) and Segelstein (1981). We use the same value for all wavelengths.

For tropospheric background aerosol, the extinction profile is inferred by averaging outside-plume observations of the lidar during the flight. Similarly, outside-plume measurements from the laser aerosol spectrometer were averaged to obtain the size distribution. For the refractive index we use a constant value of $1.53 + i \cdot 0.007$, as reported for clean continental air by Levoni et al. (1997).





**Table 2.** Overview of the data used to constrain the model for actinic flux and photolysis frequency simulations in the Shady fire plume.

| | Parameter | Data source decription | Update interval |
|---|---|---|---|
| Tropospheric background aerosol | Size distribution | Laser aerosol spectrometer measurements outside plume | Fixed |
| | Refractive index | Levoni, 1997 | Fixed |
| | Extinction profile | Lidar extinction profiles outside plume | Fixed |
| Stratospheric background aerosol | Size distribution | Wrana, 2021 | Fixed |
| | Refractive index | Levoni, 1997 | Fixed |
| | Extinction profile | SAGE II database | Fixed |
| Brown carbon (BrC) aerosol | Size distribution | Laser aerosol spectrometer | Per transect |
| | Real refractive index | Sumlin et al. (2018) | Per transect |
| | Imag. refractive index | Wavelength-dependence from spectroscopy of particle solvent extracts | Fixed |
| | | Magnitude retrieved from SSA from Nephelometer, PSAP and AOP | Per transect |
| | Extinction profile | Combined lidar backscatter profiles, Nephelometer extinction and DACOM carbon monoxide measurements | 10 s |
| Black carbon (BC) aerosol | Size distribution | Laser aerosol spectrometer | Per transect |
| | Refractive index | OPAC Hess, 1998 | Fixed |
| | Extinction profile | Same approach as for brown carbon | 10 s |
| Other | Optical black carbon fraction | Retrieved from SSA from Nephelometer, PSAP and AOP | Per transect |
| | Surface albedo | TROPOMI LER (Tilstra et al., 2023) | 10 s |
| | $O_3$ profile | Troposphere: Chemoluminescence monitor Stratosphere: Std. Atmosphere, OMI (Bhartia, 2012) column | Per transect |
| | Pressure profile | Std. atmosphere, scaled with in-situ | 10 s |
| | Temperature profile | Std. atmosphere, troposphere scaled with in-situ | 10 s |

## 4.2 Plume aerosol

In the plume, we consider the two optically dominant particle compounds: black carbon (BC) and brown carbon BrC. BC and BrC are each represented by their own aerosol type in the model. Both types use the microphysical model of the VPC aerosol module (Section 2.2.2), and we assume particles to be spherical and internally homogeneous. The results presented in this study indicate that the plume bulk optical properties can be sufficiently reproduced using this simplified approach, even though real morphology and mixing states of BB particles can be much more complex (see e.g. Hand et al., 2010; Liu et al., 2021). 400 Particle properties are assumed to be altitude-independent, as measurements performed at different altitudes with respect to the plume center did not show significant changes.



PSD parameters were inferred for each transect by averaging corresponding laser aerosol spectrometer data and fitting a bimodal log-normal distribution as accepted by the microphysical aerosol model (Section 2.2.2). We assume the same PSD for BrC and BC, since the laser aerosol spectrometer cannot distinguish between the aerosol types.

The refractive index for BC was taken from the OPAC database (Hess et al., 1998). We used a fixed value of $1.7 + i \cdot 0.46$ for all wavelengths. The real part of the BrC refractive index was set to a fixed value of $1.53$, following Sumlin et al. (2018). The imaginary refractive index $\kappa_{\mathrm{BrC}}$ was inferred from SAEB measurements. The recorded SAEB spectra (Section 3.2) reflect the wavelength-dependence of the BrC material absorption coefficient $\alpha_{\mathrm{BrC}}$ which can be converted to an unscaled imaginary refractive index using the relation $\kappa_{\mathrm{BrC}} \propto \alpha_{\mathrm{BrC}} \cdot \lambda$ (Bohren and Huffman, 1998). With the PSD and refractive index information,

we set up the microphysical aerosol model for BC and BrC mixtures to simulate bulk overall SSAs. We fit these SSAs to the ones observed by the Nephelometer, PSAP and AOP measurements to retrieve the BC to BrC ratio and the magnitude of $\kappa_{\mathrm{BrC}}$ for each plume transect. To describe the BC to BrC ratio we introduce the "optical BC fraction" which represents the contribution of BC to the total extinction of BC and BrC at a prescribed reference wavelength $\lambda_0$. It is used below to calculate separate vertical profiles for BrC and BC. Average retrieved values for $\kappa_{\mathrm{BrC}}$ and the optical BC fraction at $300\,\mathrm{nm}$ are on the

order of 0.05 and 0.08, respectively. A typical BrC imaginary refractive index obtained this way is shown in Fig. S1. The retrieved values reproduce the plume bulk optical properties well (Section 5), but it should be noted that their physical meaning is limited due to the simplifications implicit in the aerosol modeling approach.

Aerosol extinction profiles were inferred from lidar, Nephelometer, and DACOM data. Due to the limitations in the lidar aerosol extinction profiles (Section 3.2) the lidar aerosol backscatter profiles ($B$) had to be used to achieve suitable spatial reso-

lution and vertical coverage. Conversion to extinction ($E$) was achieved by investigating aerosol lidar ratios $S = E/B$ observed during plume overflights (see Fig. 3). $S$ was found to depend primarily on the smoke age $t$ (see Fig. S3). This dependence was parameterized by fitting a second order polynomial $S(t)$, used to calculate the extinction profiles $E = S(t_{\mathrm{transect}}) \cdot B$ for each transect. The vertical gap arising from the lidar's non-overlap region around the aircraft was filled using in-situ measured extinction from the Nephelometer and linear interpolation. Smaller temporal gaps in the Nephelometer data were filled using carbon

monoxide data from the DACOM instrument, applying a conversion factor inferred from Nephelometer-DACOM-correlations in the same transect (typical Pearson r correlation $> 0.99$). Missing data above or below the plume was either zero-padded or extrapolated assuming a Gaussian plume shape. Separation into BC and BrC contributions was made based on the current transect's optical BC fraction.

## 4.3 Surface reflectance

Surface reflectances are taken from the TROPOMI database for monthly Lambertian-equivalent reflectivity (LER, Tilstra et al., 2023). The database has a spatial resolution of $0.125° \times 0.125°$ ($10\,\mathrm{km} \times 14\,\mathrm{km}$ for the Shady fire location), and its 39 spectral channels cover a wavelength range from about 330 to $2300\,\mathrm{nm}$. Below $330\,\mathrm{nm}$, we assume a linear decrease to zero at $250\,\mathrm{nm}$. We consider these approximations as reasonable, since the surface albedo is generally low with minimal impact on the total actinic flux.




## 4.4 Trace gases

The only trace gas taken into account for our study was $O_3$. Based on in-situ measurements performed on the aircraft, we estimated the absorption of other trace gases to be small ($< 3\%$ in actinic flux spectra) and omitted them for simplicity. The $O_3$ tropospheric profile was inferred from $O_3$ in-situ measurements on the aircraft during ascent and descent at take-off and landing. For the stratosphere we assumed the 1976 US Standard Atmosphere, scaled such that the $O_3$ total column matched with OMI satellite observations (Bhartia, 2012).

## 5 Results

We ran the VPC model for the 20 transects indicated in Fig. 3. Simulation runs were performed at $10\,\mathrm{s}$ temporal resolution (corresponding to a horizontal spatial resolution of $\approx 1.5\,\mathrm{km}$) and 100 m vertical resolution at flight altitude, roughly matching the resolution of the lidar observations (Section 3). For each run, the model was constrained as described in Section 4, based on the plume composition and vertical distribution observed at the respective time and location. In total, 350 model runs (10 to 30 runs per transect) were performed. Each run provides vertical profiles for the actinic flux and photolysis frequencies.

For the investigation of photolysis frequencies, we will focus on four important atmospheric reactions with different spectral sensitivity:

$$O_3 + h\nu \rightarrow O_2 + O(^1D)$$
$$HONO + h\nu \rightarrow OH + NO$$
$$NO_2 + h\nu \rightarrow NO + O(^3P)$$
$$NO_3 + h\nu \rightarrow NO_2 + O(^3P)$$

Figure 4 illustrates the spectral sensitivity by showing action spectra (actinic flux spectrum $\times$ reactant absorption cross-section $\times$ quantum yield) for each reaction, assuming typical modeled actinic flux spectra with and without the plume present. The action spectra weighted average wavelengths for the four reactions are approximately 310, 360, 375, and 550 nm.

### 5.1 General features of the BB plume environment

It is instructive to start with a discussion of general features of the BB plume environment. Fig. 5 and Fig. 6 show combined overviews of modeling results and observations for four selected transects (5, 8, 14 and 20) covering different situations. Similar plots for all 20 transects can be found in Supplement S6. To provide a picture over the full wavelength range, Fig. 7 shows measured and modeled spectra for three selected model runs (as highlighted by the black rectangles in Fig. 5) with the aircraft flying within, above and below the plume.

The underlying data for the selected transects were sampled at different times of the flight (compare transect labels in Fig. 3) and at different distances to the fire (20, 40, 100 and $60\,\mathrm{km}$, respectively). As indicated by the 2D distributions of aerosol extinction coefficients (Fig. 5, A-C, and 6, A), the sampled plume cross sections have a horizontal (vertical) extent of





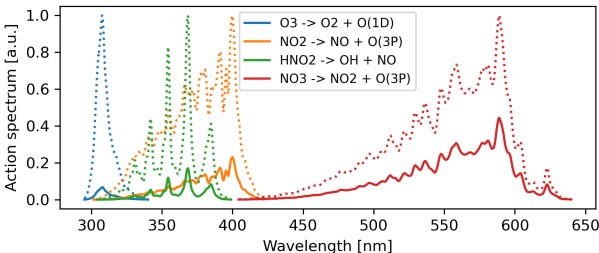

**Figure 4.** Spectral sensitivity of the four photochemical reactions considered in the comparison. Shown are action spectra (actinic flux $\times$ reactant absorption cross-section $\times$ quantum yield) assuming a typical tropospheric Rayleigh atmosphere actinic flux (dotted lines) or dense plume (AOD $\approx$ 3 at 532 nm) actinic flux spectrum (solid lines).

10 to 15 km (1 to 2 km) and are located about 2 to 3 km above the ground. Plume density and shape vary strongly between different transects. In a young and dense plume, extinction coefficients and AODs of up to 6 km$^{-1}$ and $> 3$ (both at 532 nm) are encountered, respectively (Fig. 5, A). Around the vertical middle of the plume, this leads to reductions in actinic flux (Fig. 5, G) and photolysis frequencies (Fig. 5, J) of more than an order of magnitude compared to the clean atmosphere outside the plume. The reduction is most pronounced in the UV and UV-centered photolysis frequencies, i.e. the actinic flux at 340 nm is

strongly reduced (Fig. 5, G), and is near zero at wavelengths below 340 nm (Fig. 7, A). The aerosol model results identify the increasing BrC absorption as the main reason (Fig. S1), rendering the plume darker and more opaque at the same time. BrC approximately triples plume aerosol extinction and reduces the SSA from 0.9 to 0.75, when comparing 532 to 300 nm. The radiative conditions above, within and below the plume, are very different and, depending on the wavelength, can lead to both enhancement or reduction in actinic flux and photolysis frequencies, in comparison to the situation with a clean atmosphere.

During transect 8, the relative height of the aircraft with respect to the plume varies (Fig. 5, B). We see enhancements in actinic flux and photolysis frequencies (Fig. 5, panel H and panel K), particularly towards Vis wavelengths, when the aircraft is above or in the upper region of the plume, and reduction in both UV and Vis when the aircraft moves deeper into or below the plume. This behaviour is also apparent in the example spectra (Fig. 7) as well as the actinic flux vertical profiles (Fig. 5, panels D-F and 6, panel B), and will be investigated in more detail in Section 6.

Transect 14 represents a plume older than >3 h. Older plumes are typically wider, more ragged, and optically thinner (Fig. 5, panel B, panel C). Accordingly, the reductions in actinic flux and photolysis frequencies are less pronounced (Fig. 5, panel I, panel L).

## 5.2    Comparison of model and measurement

In this section we compare the modeled actinic fluxes and photolysis frequencies to the ones observed by the CAFS. The

comparison serves to validate the VPC model and to assess the accuracy of modeling results. While the model provides full vertical profiles of actinic flux and photolysis frequencies, the comparison is limited to a single altitude per model run, namely the altitude of the aircraft, where the respective CAFS measurement was performed. However, the dataset includes



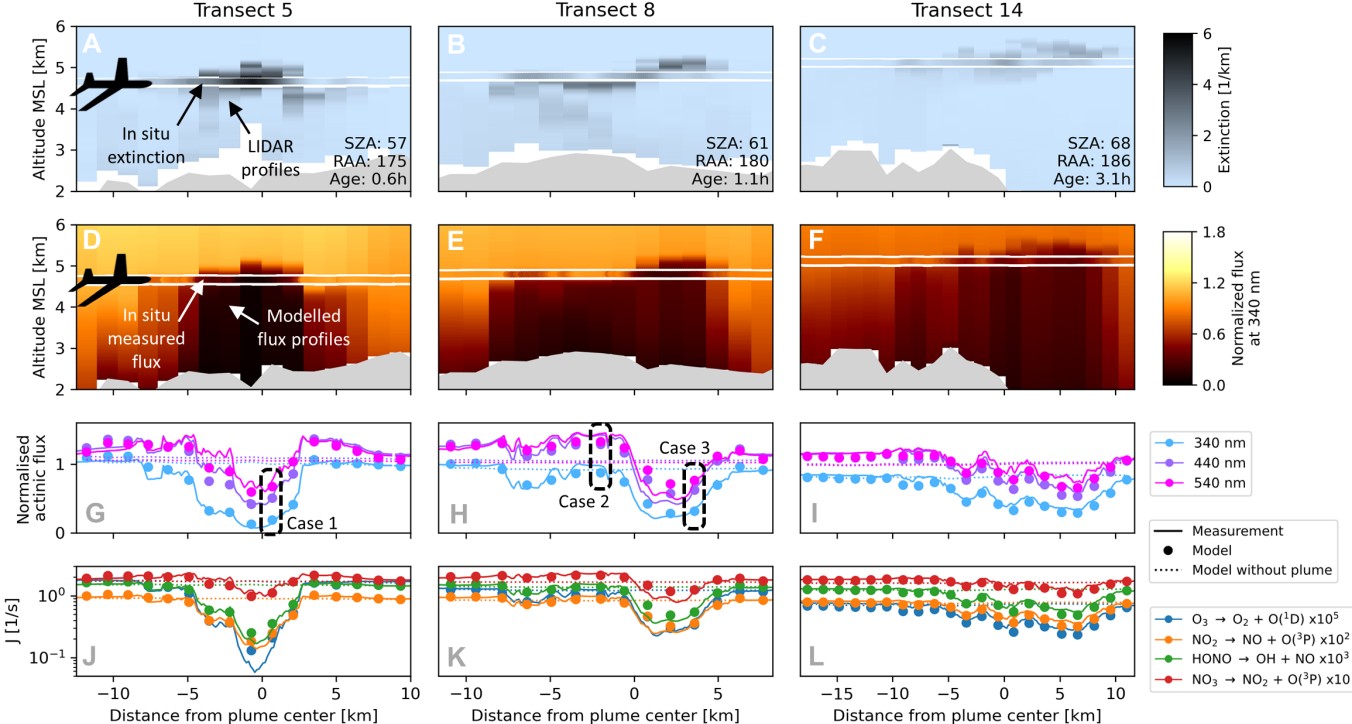

**Figure 5.** Overview of modeled and observed (CAFS) actinic fluxes and photolysis frequencies for three example transects. The first and second rows show plume cross-sections of measured aerosol extinction (532 nm) and modeled actinic flux, respectively. The observer is looking downwind. Solar relative azimuth angles (RAA) are given with respect to the viewing direction: $0°$, $90°$ or $180°$ indicate sun in front, to the right or in the back of observer, respectively. Altitudes are given with respect to mean sea level (MSL). The third row shows time series of in-situ actinic fluxes at three wavelengths, normalized to TOA irradiance. Black rectangles indicate those observations discussed in more detail throughout the paper (see also Fig. 13, Fig. 7, and Fig. 12. The fourth row shows in-situ photolysis frequencies for three example reactions. For reference, dashed lines indicate Rayleigh atmosphere actinic fluxes and photolysis frequencies, created by re-running the model without the plume present.

measurements at different relative altitudes with respect to the plume. We therefore expect the average agreement of model and measurements to be representative for the entire modeled distributions.

### 5.2.1 Comparison of actinic fluxes


The qualitative agreement of model and measurement can already be seen in the combined plots of actinic flux time series in Fig. 5, panels G-I and 6, panel C. The model reproduces major reductions and enhancements in the time series very well. Occasional outliers typically occur at the plume edges, particularly late in the evening (see highlighted model runs in Fig. 6, panel C), and are discussed later (Section 6).



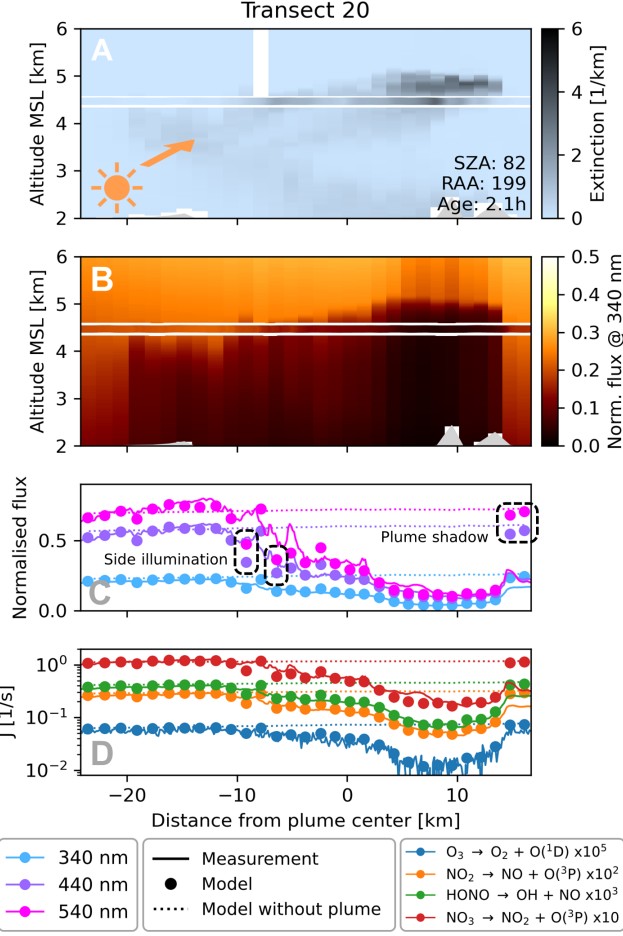

**Figure 6.** Example transect with particularly high SZA ($82°$). Legends and description of Fig. 5 apply, but note the different scales.

To quantifiy the model-measurement differences, we performed a statistical analysis. In this analysis, we compare actinic flux optical depths (AFODs), which are the logarithm of the normalized actinic flux (Eq. 6). AFODs may be less intuitive than the actinic flux itself, but their use offers several advantages: (1) AFOD root-mean-square deviations and linear regression results between model and measurements provide relative instead of absolute differences. We found relative differences to be less dependent on the actinic flux magnitude and therefore to be more representative for the full dataset. (2) Especially in dense

BB plumes a linear approach would give more weight to higher actinic fluxes, which are found outside of the plume, thus leading to improper characterization of the RT in the center of the plume. (3) Because the actinic flux varies by over an order of magnitude, a logarithmic scale will more evenly represent the agreement over the large variation of the actinic flux and is less sensitive to outliers. (4) The AFOD increases with the plume signal and for high SZAs. In this way, the linear regression offset approximately represents the model-measurement differences in clean air during daytime, and the slope reflects systematic



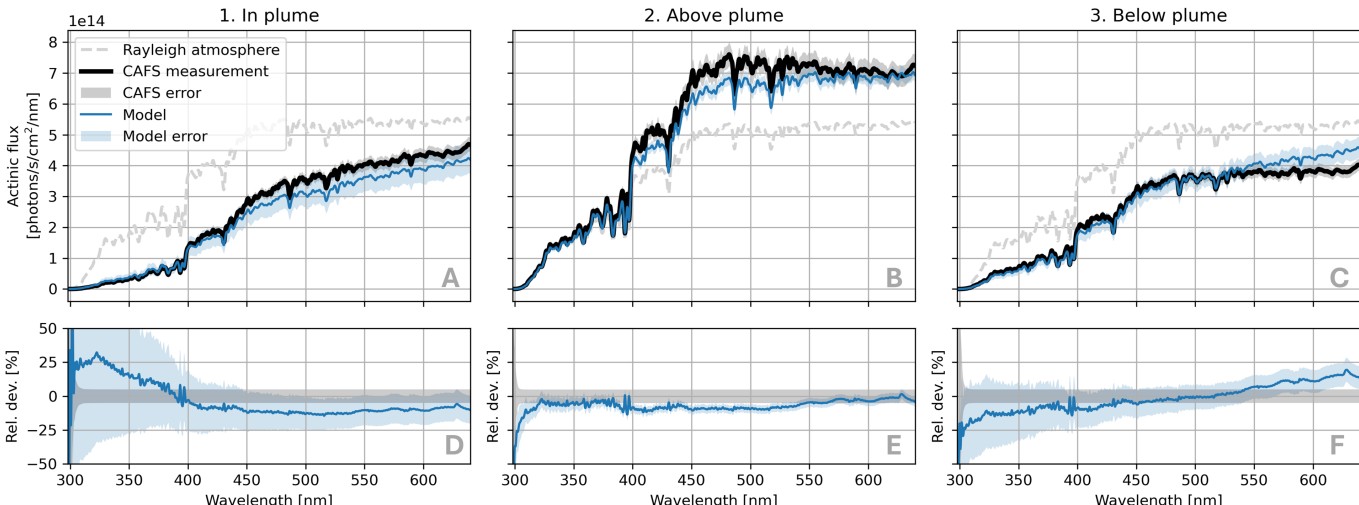

**Figure 7.** Modeled and measured (CAFS) actinic flux spectra for three example observations (as highlighted in Fig. 5). To put the flux magnitudes into perspective, "Rayleigh atmosphere" spectra were calculated by re-running the model without plume aerosols. Lower panels show relative differences between model and measurements. The modeled spectra were interpolated between simulation nodes following the procedure described in Section 2.5. Model errors were estimated using the sensitivity studies described in Section 5.2.3 and do not consider errors from horizontal radiative transport effects.

errors increasing with aerosol signal (e.g. errors in the aerosol representation or properties) and twilight conditions. This enables a more direct analysis of the RT effects inside the plume.

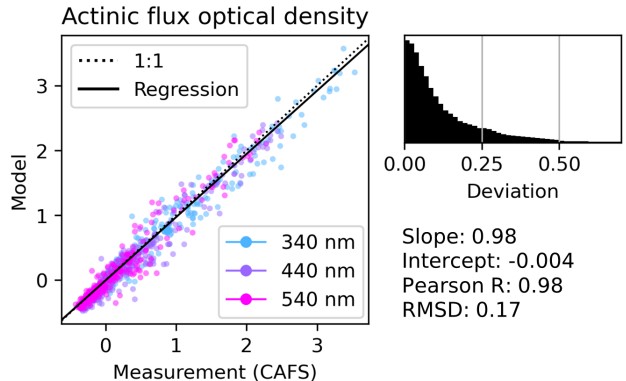

**Figure 8.** Correlation plots of measured (CAFS) and modeled (VPC) actinic flux optical density (AFOD, see Eq. 6). Data points in the plot are limited to the same wavelengths (340, 440 and 550 nm) as shown in Fig. 5. For the difference histogram and linear regression results on the right, the whole dataset was considered.

To quantify the overall agreement of model and observations, we performed a correlation analysis on the entire dataset (Fig. 8), including all 20 transects, all wavelengths, and also locations outside the plume. CAFS observations - with an original



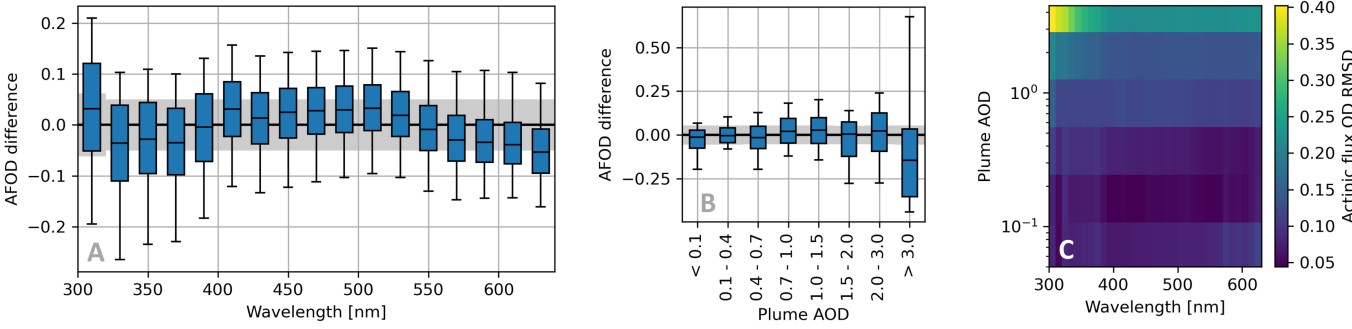

**Figure 9.** Agreement of modeled and measured actinic flux optical density (AFOD) in dependence of wavelength and plume total AOD. AODs were calculated from lidar observations at $532\,\mathrm{nm}$. Boxes span 25st to 75st percentile. Whiskers span 10th to 90th percentile. Grey shaded areas show average CAFS measurement uncertainty.

temporal resolution of $1\,\mathrm{s}$ - were averaged over each of the modeled $10\,\mathrm{s}$ time intervals. We applied two filters: (1) we only

considered data for which the CAFS signal-to-noise ratio is higher than 5 (removes $< 2\,\%$ of the data), and (2) we subsequently ignored the highest first percentile of differences to remove the most severe outliers, which are mostly associated with inhomogeneities in the plume not captured by the model. Based on this data, we obtain a Pearson correlation coefficient of 0.98 between model and measurements. The RMSD in AFOD is 0.17, which corresponds to a $17\,\%$ deviation in the actinic flux. Slope (0.98) and intercept (-0.004) from the linear regression analysis indicate no significant systematic differences. As

described in Section 2.4 and 3.3, both CAFS and VPC provide upwelling and downwelling actinic flux contributions separately. Correlation analysis results for each of the contributions are provided in Supplement S4. The upwelling part exhibits larger deviations (RMSD of $24\,\%$) than those for the downwelling part (RMSD of $17\,\%$), which is most likely due to imperfect representation of surface properties, surface illumination (e.g. shadowing by the plume) and the assumption of constant ground elevation in the model. However, the impact of this increased deviation is limited, since in the presented data, upwelling

radiation contributes on average only $\approx 20\,\%$ to the total actinic flux.

To obtain a more differentiated picture, we investigated the dependence of model-measurement differences on wavelength, plume AOD and solar geometry by binning of the data (Fig. 9 and 10). For the AOD binning, we used the AODs at $532\,\mathrm{nm}$ observed by the lidar. Overall, we found this AOD to be a good proxy to identify those observations significantly affected by the presence of the plume, even though the impact of the plume signal also depends on the aircraft's altitude with respect

to the plume. We find a clear increase in model-measurement difference towards short wavelengths (Fig. 9, A and also Fig. 7) and high AODs (Fig. 9, B). For wavelengths $> 400\,\mathrm{nm}$ and low AODs, RMSDs of 0.05 to 0.1 are observed. In contrast, for wavelengths around $\approx 300\,\mathrm{nm}$ and high AODs ($\approx 4$ at $532\,\mathrm{nm}$ and $\approx 10$ at $300\,\mathrm{nm}$), RMSDs increase up to 0.4 (Fig. 9, C). At high AODs, Fig. 9 B indicates a general systematic underestimation. Only observations above and in the upper plume contribute to this data bin, since observations below do not meet the CAFS signal-to-noise filtering criterion mentioned before.



For intermediate AODs ($1 <$ AOD $< 3$ at $532\,\text{nm}$), including all wavelengths, the RMSD is 0.18. We consider this value to be representative for the typical in-plume model accuracy.

For the dependence on the solar geometry, we chose three segments of the flight (transects 1, 2 and 3; transects 6, 7 and 8; transects 18, 19, 20). ) with different solar geometries but otherwise similar conditions (Fig. 10, A). During the first and third segment, the Sun illuminates the plume from the side (albeit at a small relative azimuth angle of $\approx 15°$) with respect to the

plume axis), at low ($\approx 50°$) and high SZA ($\approx 80°$), respectively. During the second segment, the sun is approximately aligned with the plume axis at a moderate SZA ($\approx 60°$). During side illumination, in particular at high SZAs, model-measurement differences increase and model results underestimate actinic fluxes (Fig. 10, B). Compared to the second segment, the RMSD for the third segment almost doubles (from 0.13 to 0.24). As discussed in detail in Section 6, this might be caused by the 1D model assumption of a horizontally homogeneous atmosphere.

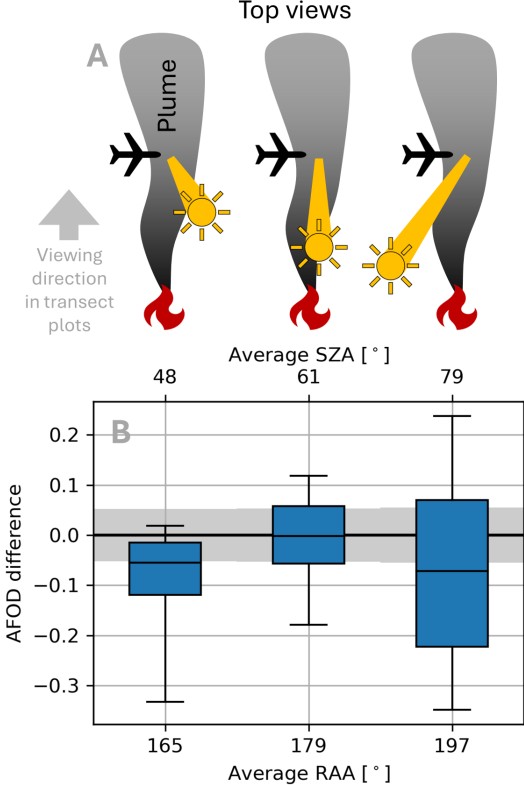

**Figure 10.** Agreement of model and measurement for three segments of the measurement flight with different solar geometries, as illustrated in Panel A. RAAs were calculated with respect to the plume axis, i. e. the prevalent wind direction. Boxes span $25^{\text{th}}$ to $75^{\text{th}}$ percentile. Whiskers span $10^{\text{th}}$ to $90^{\text{th}}$ percentile. Grey shaded areas show average CAFS measurement uncertainty.





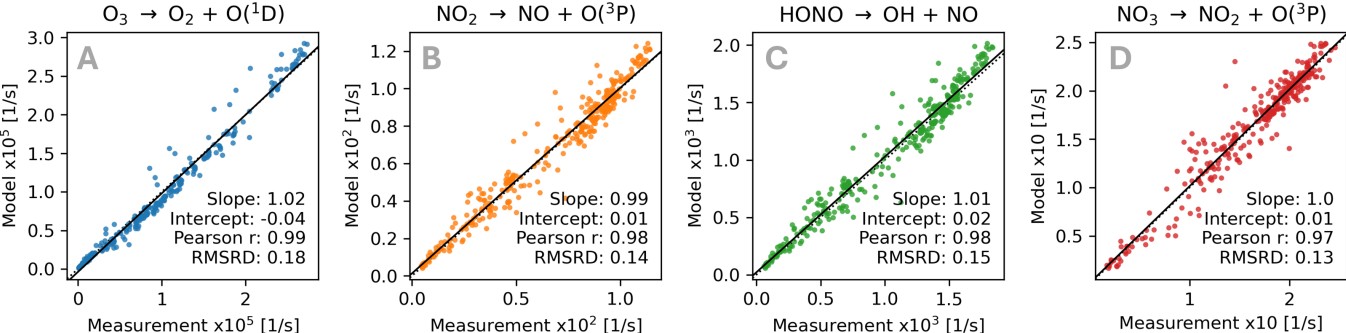

**Figure 11.** Agreement of modeled and measured photolysis frequencies for the four example reactions with different spectral sensitivity. Instead of root-mean-squares of absolute differences, we show root-mean-square values of relative differences here (RMSRD), for easier comparison of different reactions.

### 5.2.2 Comparison of photolysis frequencies

From a chemical point of view, it is interesting to asses how the model-measurement differences in actinic fluxes propagate into the photolysis frequencies. We therefore performed correlation and regression analyses similar to those in Fig. 8 for the modeled and measured photolysis frequencies of the 48 tropospheric photochemical reactions already mentioned in Section 2.4. Again, the first percentile of data with highest differences was omitted. Results for the four reactions introduced at the beginning of Section 5 are shown in Fig. 11. The results for the remaining reactions are listed in Supplement S2. Note that, instead of the root-mean-square of absolute deviations (RMSDs), we calculate the root-mean-square of relative deviations (RMSRD) here, according to:

$$\text{RMSRD} = \sqrt{\frac{1}{N}\sum_{i=1}^{N}\left(\frac{y_i - x_i}{x_i}\right)^2} \tag{9}$$

with $x_i$ and $y_i$ being measured and observed photolysis frequencies, respectively. This facilitates the comparison between different reactions.

As mentioned before, CAFS and VPC use the same photolysis frequency code, quantum yield and absorption cross-section data. The conversion from actinic fluxes to photolysis frequencies therefore does not introduce additional differences between model and measurement; photolysis frequencies simply reflect the actinic flux differences, weighted by the action spectrum of the corresponding reaction (Fig. 4). Accordingly, systematic differences (slope and offset of the linear regressions) and RMSRDs for the photolysis frequencies are similar to those obtained for actinic fluxes and follow the patterns already seen in panel A of Fig. 9. Larger differences are found for UV-driven reactions like the $O_3$ photolysis (Fig. 11, A). Smaller differences are observed for Vis-driven reactions, especially the $NO_3$ photolysis (Fig. 11, D), which also benefits from integration over a broad spectral range (see action spectrum in Fig. 4). For all 48 reactions, Pearson correlation coefficients are $> 0.96$, slopes





are between 0.96 and 1.04 and intercepts range between $-5\%$ and $+2\%$ of the average observed photolysis frequency for the
respective reaction (Supplement S2). RMSRDs are between 0.12 and 0.21, with an average of 0.15.

### 5.2.3   Model sensitivity to input parameters

To better understand the origin of the model-measurement differences and identify critical factors, we performed model sensitivity studies, i.e. we investigated the response of the modeling results to variations in the input parameters.

**Table 3.** Parameter variation magnitudes

| Group | Parameter | Variation |
|---|---|---|
| PSD | Fine mode median radius $r_1$ | 10% |
| | Coarse mode median radius $r_2$ | 10% |
| | Fine mode width $\sigma_1$ | 0.1 |
| | Coarse mode width $\sigma_2$ | 0.1 |
| | Modal fraction $f$ | 0.1 |
| Refractive index | Real part $n$ | 0.1 |
| | Imaginary part $\kappa$ | 20% |
| Other | Ozone column | 5% |
| | Surface LER | 100% |
| | Aerosol extinction | 10% |
| | Temperature | 5% |
| | Pressure | 2% |
| | Optical BC fraction | 0.05 |

The variations applied to the model inputs are listed in Table 3. Their magnitudes correspond to the approximate uncertainties
in the respective parameter. Variations of PSD parameters, pressure and temperature were estimated based on specifications of
the corresponding FIREX-AQ instruments, but also considering noise, variability and limited coverage of the measurements.
Refractive index and optical BC fraction variations are based on the variability reported in the literature (Sarpong et al.,
2020; Lack et al., 2012; Andreae and Gelencser, 2006) and from uncertainties in SSA measurements, propagated through
the Mie model fit described in Section 4.2. $O_3$ column and surface LER variations are based on reported uncertainties in
the corresponding satellite measurements (Shavrina et al., 2007; Balis et al., 2007; Kroon et al., 2008; Frith et al., 2020;
Tilstra et al., 2023). Aerosol extinction uncertainty was assumed to be dominated by uncertainties in the lidar ratio, which we
calculated from the scattering in Fig. S3. The listed variations were applied to the three simulated cases highlighted in Fig. 5
and shown in Fig. 7.

The resulting differences in the modeling results (Fig. 12), including their spectral dependencies, exhibit very similar magni-
tudes and patterns as those differences between model and measurements reported in Section 5.2.1. Most parameters induce a



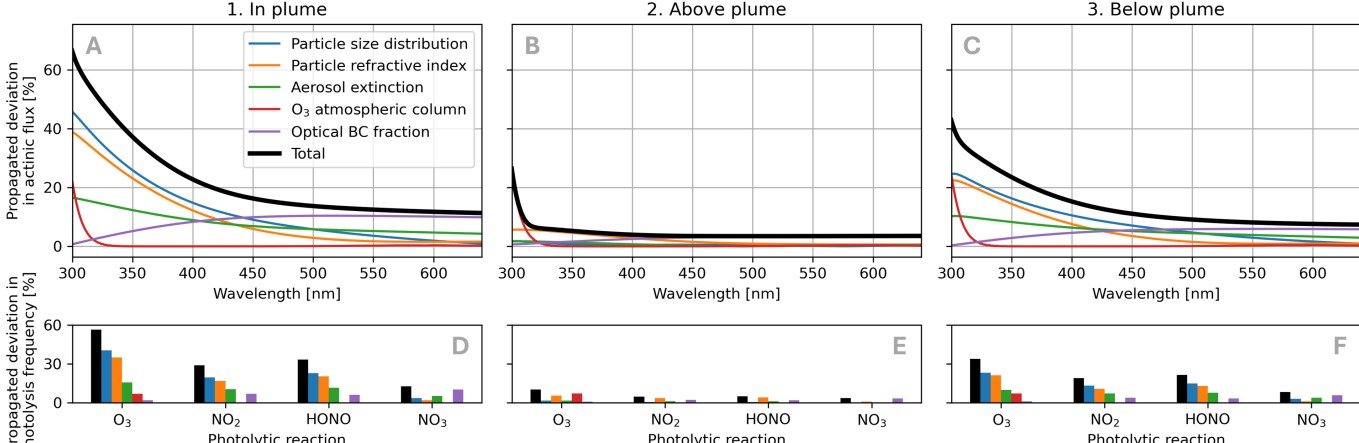

**Figure 12.** Response of actinic flux spectra (panel A, B and C) and photolysis frequencies (panel D, E and F) to the parameter variations defined in Table 3, for the three example spectra presented in Fig. 7. Parameters have been grouped here for readability by quadratically adding their propagated impact. Legend in panel A applies to the entire figure.

spectrally smooth difference in the actinic flux, which increases significantly towards the UV (Fig. 12, A and B). Furthermore, the differences increase with the plume AOD, i.e. they are largest (with a total of up to $60\%$ in the UV) inside the optically dense plume of transect 5 (Fig. 12, A), and become smaller (up to $40\%$) for the thinner plume encountered during transect 8 (Fig. 12, C). Smallest differences are found above the plume (Fig. 12, B). Here, the actinic flux is dominated by the contri-
bution from direct solar radiation (see also Fig. 13), which is not affected by plume aerosols, but prone to uncertainties in the stratospheric $O_3$ column below $315\,nm$.

In and below the plume, uncertainties in the UV are dominated by particle properties such as refractive indices and the PSD. Investigating the contributions of separate parameters and aerosol types (not shown in Fig. 12) provides further insights: as expected from the generally low optical BC fraction ($10\%$ at around $500\,nm$) and the strong increase of BrC absorption towards
the UV, these uncertainties are dominated by the BrC properties, while BC parameter variations contribute less than $20\%$ to the total uncertainty below $350\,nm$.

In the Vis, uncertainties are generally lower than those in the UV, since plume optical thickness decreases with wavelength. The uncertainty is dominated by the optical BC fraction, which is approximately proportional to the plume extinction. The contribution from BrC becomes negligible towards longer wavelengths.

**6    Discussion**

The major features in the observed actinic flux and photolysis frequency time series (Fig. 5 and 6), as well as in the actinic flux spectra (Fig. 7), are well reproduced by the model. RMSDs over the entire dataset are on the order of $10\%$ to $20\%$, in both actinic fluxes and photolysis frequencies. Systematic differences in our comparison are on the order of a few percent, as



illustrated by the regression slope of 0.98 and negligible intercept between measured and modeled actinic fluxes (Fig. 8) and

slopes between 0.99 to 1.02 in the measured and modeled photolysis rates (Fig. 11). These differences are smaller than the uncertainty in the CAFS measurements (Fig. 8, 9, 11).

Generally we find a surprisingly good model-measurement agreement, considering the complexity, heterogeneity and variability of BB plumes and the large variations in actinic flux and photolysis frequencies over more than an order of magnitude on short spatial and temporal scales. To put our results into perspective: similar airborne (Kelley et al., 1995; Volz-Thomas

et al., 1996) and ground-based (Barnard et al., 2004; Castro et al., 1997; Dickerson et al., 1997; Kazantzidis et al., 2001; Vuilleumier et al., 2001; Balis et al., 2002; Shetter et al., 2003; Hofzumahaus et al., 2004) model-measurement comparisons for less polluted and nearly homogeneous clear-sky atmospheres typically achieve agreements on the order of $10\%$.

Besides the relatively small CAFS measurement error, model-measurement differences arise for two major reasons: Errors in the model input parameters and simplifications taken in the modeling approach.

Errors in model input parameters partly arise from instrumental measurement errors and uncertainties in the literature data that were used to constrain the model (Section 4). Furthermore, the limited spatio-temporal coverage and resolution of this data does not fully encompass real variations in the inhomogeneous plume, thereby adding considerable uncertainty to the model constraints. Non-linear relations between plume properties and actinic fluxes might also hinder exact simulations when using averaged parameters.

Model simplifications are the representation of aerosol (as spherical particulates and externally mixed) and the application of a 1D model, which cannot account for horizontal RT effects. The latter are caused by the horizontal inhomogeneities of the scenario. Such effects comprise for instance side-illumination and shadowing, especially at the edges of the plumes. But also on smaller scales and within the plume, horizontal RT can affect the results. For instance, the fact that the lidar aerosol extinction profile was measured in zenith direction while the RT calculation were performed for non-zero SZA can introduce

inconsistencies in the aerosol extinction profiles.

For our sensitivity studies (Section 5.2.3), we estimated the uncertainties in the model input parameters, considering both instrumental errors and uncertainties due to limited coverage and resolution of the measurements. Propagating these uncertainties through the model, yields model uncertainties very similar to the observed model-measurement differences, not only in magnitude but also in the wavelength and AOD dependence (compare Fig. 9 and 12). We conclude that, at least as average

behavior over the dataset, the model-measurement differences are dominated by uncertainties in the model input. Hence, even with comprehensive state-of-the-art measurements as performed during FIREX-AQ, the information on the plume environment is still the limiting factor for accurate modeling of actinic fluxes and photolysis frequencies in BB plumes. Most critical is the uncertainty in brown carbon aerosol properties (refractive indices and PSD), which dominates the model uncertainties in the UV (Fig. 12), where photochemistry is most sensitive. BrC properties are difficult to assess as they are highly variable and the

mechanisms affecting the absorption's spectral dependence are not yet fully understood (e.g. Laskin et al., 2015; Shetty et al., 2023). A recent comparison study revealed inconsistencies between different approaches to derive BrC optical properties from FIREX-AQ data (Zeng et al., 2022), presumably in part due to insoluble BrC components that the SAEB technique (Section 3) cannot detect (Liu et al., 2013; Shetty et al., 2019; Chakrabarty et al., 2023). In our study, this effect is likely reduced,



since we combine the SAEB observations with direct in-situ optical measurements of the SSA (Section 4.2). However, the absence of SSA observations in the UV and the inhomogeneity of the plume hinders an accurate consideration of BrC in the modeling process. We propose that further investigation of the discrepancies between in-situ extractive BrC sampling and remotely-sensed actinic flux or radiance spectra may shed light on this issue and ultimately better constrain radiative transport and photochemistry in BB plumes.

Considering that most of the model-measurement differences can be explained by uncertainties in the model input, simplifications in the model are unlikely to limit the modeling accuracy under typical conditions. However, horizontal radiative transport effects might become problematic for specific cases. These effects - in particular side-illumination and shadowing - are expected to be most pronounced when the SAA is not aligned with the plume axis, i.e. the Sun illuminates the plume from the side (first and third sketch in Fig. 10, A) and when SZAs are large. Indeed, we observe a dependence in model-measurement differences, matching these expectations (Fig. 10, B). Furthermore, horizontal radiative transport effects are likely to occur at the plume edges, where horizontal inhomogeneity is large. An extreme case, where all three conditions are fulfilled, are the plume edges of transect 20. The actinic flux time series indicates strong under- and over-estimations (black boxes in Fig. 6, C), which can be explained by plume side-illumination and shadowing, respectively. It should be noted that the Shady Fire dataset is favourable in this context, since the sun is almost aligned with the plume axis over the entire flight (Section 3). Plumes under less favourable geometries might lead to larger horizontal radiative transport effects. On the other hand, corrections, for example based on additional measurements of direct solar AOD, might reduce these effects in the future (Várnai and Davies, 1999).

With the confidence that we can model actinic fluxes and photolysis frequencies accurately, it is worth discussing the modeled spatial distributions of actinic fluxes and photolysis frequencies. Figure 13 shows actinic flux vertical profiles for Case 1 during transect 5 (black box in Fig. 5, panel G). We separated contributions from direct, diffuse downwelling and diffuse upwelling flux and investigated two wavelengths close to the lower and upper end of the photochemically relevant wavelength range (320 and 600 nm, in panel A and panel B, respectively). For the wavelengths in between these values, the profiles were found to transition steadily into each other.

Below the plume, the contribution from direct sunlight is considerably reduced in the UV and Vis, leading to a net reduction in the actinic fluxes, despite the enhancement of the diffusive radiation fields (e.g. the Vis downwelling contribution in Fig. 13, panel B). Above the plume, the behaviour is very different for UV and Vis. In the Vis (Fig. 13, panel B), more light is scattered upwards, as the plume albedo is larger than the typically dark Earth surface (LER $\lesssim 10\%$). This leads to an enhancement of the Vis diffuse upwelling flux on the order of $20\%$ for the example in Fig. 13, panel B, and up to $60\%$ over other parts of the Shady fire. The enhancement decreases only slightly with altitude for the infinitely horizontally extended plume in our 1D model. This behaviour is very similar to the one reported for cloud layers, which are known to have substantially ($> 100\%$) increased (decreased) actinic fluxes and photolyis frequencies above (below) the clouds (e.g. van Weele and Duynkerke, 1993; Lefer et al., 2003).

This behavior changes significantly in the UV. While upwelling radiation from clouds is similar in the UV and Vis spectral range, actinic fluxes above BB plumes remain similar to those in a pure Rayleigh atmosphere. This is due to increased darkening



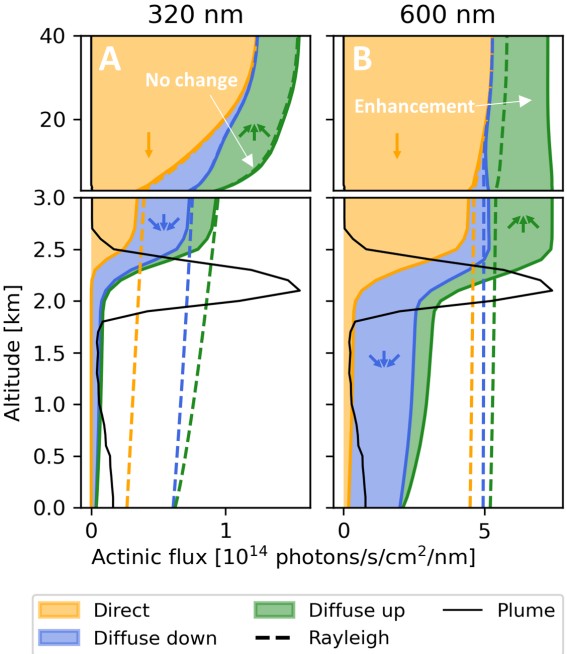

**Figure 13.** Detailed plot of the simulated actinic flux vertical profile, for the case 1 in transect 5 (black box in Fig. 5, panel G), in the UV (left) and Vis (right). Colored areas indicate separate contributions from direct as well as diffuse upwelling and downwelling radiation. The thin black line indicates the plume extinction profile in arbitrary units. Note the different altitude scales of upper and lower panels. To put fluxes into perspective, dashed lines show results for a Rayleigh atmosphere without plume.

of the plume by BrC absorption. For wavelengths below 320 nm, actinic fluxes above the plume are even reduced by a few
665 percent (light gray shading above the plume in Fig. 14, panel B), indicating that the plume appears even darker than the combined reflectance of surface and atmosphere in a clean atmosphere. Within and below the plume the shape of the profile is similar for UV and Vis but reductions are much stronger in the UV.

For chemical studies, the spatial distribution of photolysis frequencies in the plume is of most interest. Figure 14 shows such distributions, as obtained from the VPC model based on data from transect 5. The distributions mostly reflect the findings on
670 the actinic flux vertical profiles. Photolysis frequencies for UV-driven reactions such as the $O_3$ photolysis (Fig. 14, B) remain unchanged (or are slightly lower) above the plume, and drop to near-zero towards the center or bottom of the plume. For Vis-driven reactions like the $NO_3$ photolysis (Fig. 14, D), photolysis frequencies are enhanced by up to $40\%$ above the plume, while the reduction within and below the plume by up to $60\%$ is much weaker than that in the UV. The ultimate impact of these spatial and spectral features on the plume processing should be investigated with high resolution chemical models in the
675 future.



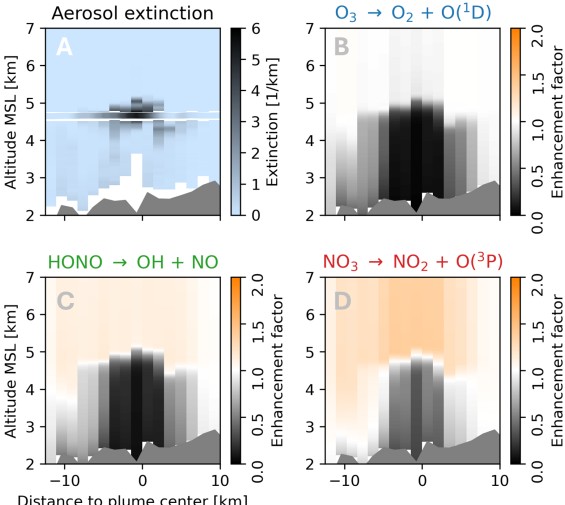

**Figure 14.** 2D distributions of three photolysis frequencies (Panel B,C,D) for the transect 5 plume cross-section. To illustrate the situation, panel A shows the plume shape in terms of aerosol extinction (same as Fig. 5, panel A). "Enhancement factors" represent the ratio of modeled photolysis frequencies with and without plume.

## 7 Conclusions

We introduced and validated VPC, a VLIDORT-based quasi-spherical 1D RT model optimized for RT and remote sensing applications in BB plumes. The model, which is constrained by a comprehensive set of aerosol measurements performed during FIREX-AQ, calculates actinic flux and photolysis frequencies in BB plumes with an accuracy of $10 - 20\%$ compared to direct measurements.

The average difference of actinic fluxes over the entire dataset is $17\%$, with larger values ($\approx 40\%$) at $\approx 300\,\text{nm}$ and for large plume AODs ($\approx 10$ at $300\,\text{nm}$). For the resulting photolysis frequencies, RMSRDs range between $13\%$ and $21\%$, with lower (higher) errors for Vis- (UV-) driven reactions. A further analysis of the comparison results show very small systematic differences on the order of $\approx \pm 2\%$ between the model and the observations. Our sensitivity studies suggest that most of the RMSD can be explained by the uncertainties in the model input data, mostly arising from the limited spatio-temporal coverage and resolution of the corresponding measurements. In the UV, the model error is dominated by the uncertainty in the properties of the strongly absorbing BrC. These uncertainties can stem from the variability of refractive indexes as well as size distribution inside the plume. In the Vis, the amount and properties of black carbon dominate the model error. Due to the decrease of both strong BrC absorption and plume optical density towards the Vis, the model error in the Vis is lower by about a factor of 3 compared to the error in the UV. Given these findings, and considering that photochemistry is particularly sensitive to the UV spectral range, more research on BrC optical properties in BB plumes is needed to better constrain radiative transport and photochemistry in BB plumes.



The model-measurement comparison also provides insights into the potential limitations of the 1D approximation in the RT simulations. We observe a systematic increase in the model-measurement difference by a factor of two for solar geometries with high SZA and when the SAA is not aligned with the plume direction. In addition, occasional outliers at the plume edges, where horizontal gradients of environmental parameters are largest, can be identified. Both these considerations indicate that, under certain conditions, 3D RT effects can have significant impact.

Previous model-measurement comparisons in clean atmospheres have found differences of $< 10\%$. Considering the highly complex and inhomogeneous RT environment in dense BB plumes, the agreement between VPC and the observations is remarkably good.

The model results also provided insights on how photochemistry is affected within and in proximity of BB plumes. Despite the absorption of BC, BB plumes appear bright in the nadir at visible wavelength compared to the typical Earth surface. Accordingly, the additional upward reflected light leads to enhancements of actinic fluxes in the Vis above the BB plume of up to $\approx 60\%$ compared to fluxes in a Rayleigh atmosphere. This enhancement can reach the upper troposphere and the stratosphere, and resembles the impact of clouds on actinic fluxes, reported in previous studies. However, the increase in BrC absorption significantly darkens the BB plume towards the UV. Consequently, the observed above-plume enhancement gradually disappears towards shorter wavelengths and actinic fluxes can even be reduced compared to fluxes in a Rayleigh atmosphere for wavelengths $\lesssim 310\,\text{nm}$. Inside the plume, actinic fluxes steadily decrease from top to bottom and remain approximately constant in the atmosphere below. The decrease is more pronounced in the UV than in the Vis.

Our results show very good agreement between observations and measurements in the Shady fire. However, our results also highlight the challenges in describing actinic fluxes in complex dense BB plumes. The Shady fire was selected for its large plume size, clear sky conditions during the flight, and the alignment of the sun with the plume axis which reduce plume inhomogeneities and 3D radiative transport effects. An expansion of our analysis to the entire FIREX-AQ dataset of over 90 fires would allow assessment of actinic fluxes and RT modeling challenges in a wider variety of biomass burning (BB) plumes. Identifying the most critical unknowns from additional sensitivity studies might be of great value in the future to constrain actinic fluxes in and around BB plumes with less measurements and efforts.

The availability of a linearized RT model, such as VLIDORT-QS, together with our sensitivity results also open the opportunity to use actinic flux and other remote sensing observations to study aerosol optical properties. The increasing disagreement of measured and actinic fluxes towards lower wavelength strongly implies that the BrC optical properties from the in-situ SAEB observations may not fully represent the true properties of the plume aerosol. The next application for VLIDORT-QS is to retrieve BrC aerosol optical properties from the measured actinic flux spectra.

Adding the facility to calculate actinic fluxes to VLIDORT-QS allows us to improve trace gas remote sensing retrievals of BB plumes. These retrievals depend on the trace gas vertical concentrations profiles, which for many of the target gases, such as $O_3$, $NO_2$, HCHO, and HONO, depend on the vertical profiles of the respective photolysis frequencies. By combining actinic flux profiles from VLIDORT-QS with a chemical transport model, self-consistent retrievals of trace gases should be possible.



**Table 1.** Abbreviations

| | |
|---|---|
| AFOD | Actinic flux optical depth |
| AGL | Above ground level |
| AOD | Aerosol optical depth |
| BB | Biomass burning |
| BC | Black carbon |
| BrC | Brown carbon |
| CAFS | Charged-coupled device Actinic Flux Spectro-radiometer |
| MSL | Mean sea level |
| PSD | Particle size distribution |
| RAA | Solar relative azimuth angle |
| RMSD | Root-mean-square deviation |
| RMSRD | Root-mean-square of relative deviation |
| RT | Radiative transfer |
| SAA | Solar azimuth angle |
| SAEB | Spectral analysis of extracted BrC chromophores |
| SSA | Single scattering albedo |
| SZA | Solar zenith angle |
| UV | Ultra-violet (spectral range) |
| Vis | Visible (spectral range) |
| VPC | VLIDORT for photochemistry |

*Code and data availability.* The raw data from the FIREX-AQ campaign is available on the "NASA Airborne Science Data for Atmospheric Composition" database: https://www-air.larc.nasa.gov/cgi-bin/ArcView/firexaq. The processed model input data and simulation results are available at DOI 10.5281/zenodo.12802619. The latest version of the VPC model is on a private Github repository. Access is provided by the authors on request.

730  1

*Author contributions.* JLT performed the RT simulations, conducted the comparison and wrote the first draft of the manuscript. JLT, SFC and NB developed the VPC framework. RS and MC expanded the VLIDORT-QS Fortran code and provided support for its use in the VPC framework. SH and KU collected and processed the CAFS data and advised on its assimilation in the study. JH and TS collected and processed the lidar profiles and advised on their assimilation in the study. RW and JD collected and processed the SAEB data and advised on
its assimilation in the study. RM collected and processed data from the TSI Laser Aerosol Spectrometer, the PSAP and the Nephelometer. VN and NT provided advice on RT, aerosol property and satellite remote-sensing aspects of the work. JS initiated the investigations and contributed to all project activities as the primary supervisor. All authors contributed to the manuscript.



*Competing interests.* The authors declare that they have no competing interests.

*Acknowledgements.* UCLA's research was funded by NOAA Grant: NA17OAR4310005 (Remote Sensing of Radical Precursor Chemistry
in Biomass Burning Plumes) and, NASA Grant number 80NSSC21K1447 (Advancing UV/Vis remote sensing of biomass burning plumes:
Brown carbon, actinic flux, and trace gas retrievals). RW was supported by NASA award No. 80NSSC18K0662. JD was supported by
NASA grant 80NSSC18K0631. SH and KU are supported by the NSF National Center for Atmospheric Research, which is a major facility
sponsored by the U.S. National Science Foundation under Cooperative Agreement No. 1852977. Their research was funded by NASA Award
Nos. 80NSSC18K0638 and 80NSSC21K1446. Part of this research was carried out at the Jet Propulsion Laboratory, California Institute of
Technology, under a contract with NASA (grant No. 80NM0018D0004). We acknowledge the following FIREX-AQ PIs and participants
for providing their data: Glenn Diskin (DACOM), Nick Wagner (NOAA AOP) and Elizabeth Wiggins (NASA LARGE). We acknowledge
the free use of the TROPOMI surface DLER database provided through the Sentinel-5p+ Innovation project of the European Space Agency
(ESA). The TROPOMI surface DLER database was created by the Royal Netherlands Meteorological Institute (KNMI).



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
