# Peer review of "Modeling actinic flux and photolysis frequencies in dense biomass burning plumes"

_EGUsphere, 2024_

## Author Comment (AC1)

**Introduction**

We would like to thank the reviewers for their valuable and constructive feedback!

The initial reviewer comments are printed in **bold**, our answers in regular font.

All line numbers refer to the initially submitted manuscript.

**Review 1: Specific comments and answers**

**Line 52. In addition to spatio-temporal coverage, such measurements are often not collected on all trace gas or aerosol aircraft missions which limits the amount of data and types of conditions that are available to evaluate models. This point might be useful to include.**

We extended the paragraph accordingly to: *"However, such in-situ measurements are limited in spatio-temporal coverage and are not performed on all aircraft missions studying trace gases or aerosols."*

**Line 80: A similar issue arises for high resolution cloud simulations where 3-D radiative effects become important and have been observed near the edges of clouds. If the resolution of a model is coarse, the 3-D radiative effects become subgrid scale processes that are often ignored (and may be small anyway).**

We included this information by extending the corresponding paragraph from its original wording:

*"Simulation of horizontal radiative transport effects, such as side illumination and shadowing require three-dimensional RT models (e.g. Mayer, 2009; Deutschmann et al., 2011)."*

To:

*"In the presence of optically thick clouds or plumes, horizontal transport effects such as side illumination and shadowing can significantly impact the radiation field, particularly when spatial scales on the order of the cloud size or smaller have to be resolved (Stephens and Platt, 1987; Mayer, 2009; Wagner, 2023). Considering such effects requires three-dimensional RT models (e.g. Mayer, 2009; Deutschmann et al., 2011), which are computationally expensive."*

**Line 95-101: What is missing here is that validating 1-D RT models, such as the one in this study, using the best input data is needed to understand uncertainties introduced into parameterization of 3-D model predictions. The 3-D model predictions are not constrained, so that errors can come from many sources. If the predicted actinic fluxes are not correct in a 3-D model simulation, most likely the**

**errors arise from simulated aerosol number, mass, composition and assumptions in optical parameters (e.g. brown carbon), rather than from the 1-D RT model formulation.**

We agree with this sentiment and added a corresponding paragraph. However, since 3D chemical transport models are not in the main scope of the publication and we cannot provide a satisfying answer to this question at this point, we added it as outlook to the conclusions (L722):

*"Further applications are conceivable in the context of chemical transport modeling (CTM). To improve computational efficiency, CTMs typically run at relatively coarse spatial resolution (several km) and resort to simplifications (e.g. 1D-RT modeling) and parametrisations to account for photochemical processes. Detailed modeling based on accurate measurements of the atmospheric state, as presented in this study, can help to understand magnitudes and origins of the uncertainties introduced into CTMs by these approximations."*

**Line 126: What type of aerosol model is assumed? Or does it matter? Since this is an off-line calculation, it is just about specifying input? To many readers an "aerosol model" implies some sort of prognostic treatment of aerosols, so how that phrase is used can be confusing. Bulk, modal, and sectional models treat the aerosol size distribution differently, and models have different representations of mixing state that affect aerosol optical properties. Some additional discussion is needed. It looks like some other discussion is included in Section 2.2.2, but think this should be mentioned here as well.**

We would like to point out that the wording is "aerosol module", not "aerosol model" here. Nevertheless, we agree that adding more information at this point is helpful to prevent misunderstanding. We changed the paragraph:

*"On the input side, VPC features the Aerosol module, which can account for multiple types of aerosol in a flexible way (Section 2.2.2)."*

To:

*"On the input side, VPC features the so-called "aerosol module", designed to describe complex aerosol mixtures in a flexible way: the module allows usage of an arbitrary number of aerosol types, each with individual properties that are described either via bulk optical or microphysical parameters (see Section 2.2.2 for details)."*

**Line 171: Since particles not are treated as coated as an option, should one assume the aerosol composition is treated as an internal mixture?**

Not necessarily. For clarification we extended the paragraph:

*"In the current VPC version, the Mie model assumes homogenous particles, i.e. without coatings. Internally or externally mixed aerosols are realized by defining a single internally mixed aerosol type or multiple externally mixed aerosol types with individual properties, respectively. In the presented study we make use of the latter approach (see Section 4)."*

We further corrected the following sentences:

L160: *"... by a single set of effective bulk aerosol optical properties ..."*

L164: *"... available aerosol models, provides aerosol properties as ..."*

**Lines 325-332: Is it important to account for ambient aerosol water for this case? If so, how was that done. It looks like measurements were made for low RH conditions that may differ from ambient conditions. Neglecting aerosol water (if present) would adversely affect aerosol optical property calculations.**

We believe that aerosol water it is not of importance for the studied case, because the air was dry, and because corresponding investigations did not indicate significant hygroscopic growth effects. For clarification, we extended the corresponding paragraph by:

*"The average relative humidity for the investigated flight segments was (32+/-5) %. Hygroscopic scattering enhancements for BB aerosol at such humidities are reported to be on the order of 1% and below (Kotchenruther and Hobbs, 1998; Chang et al., 2023)."*

And by:

*"A comparison of data for dry and ambient conditions did not indicate hygroscopic growth effects exceeding other measurement uncertainties."*

**Line 381: It is understandable to average the measurements over some period to reduce noise, but the authors average some variables and not others. Why not average all of them?**

Our goal was to perform simulations at the highest temporal resolution possible (ultimately limited by the 10s resolution of the Lidar). The decision regarding the averaging interval of each variable was taken considering:

1. Is there high temporal resolution data available at all?
2. Is the in-transect variability of the measured quantity and its impact on the radiative transport expected to be negligible compared to measurement noise/uncertainties?

To make this clearer we extended the corresponding paragraph:

*"As indicated in Table 2, different model input parameters are updated at different temporal intervals, depending on the parameter's data availability, variability and its relevance for the RT:*

*1. 10 seconds: the temporal resolution of the model simulations is ultimately limited by the resolution of the lidar backscatter profile measurements (10s). All model input data available at shorter time spans are therefore averaged to at least 10s intervals prior to simulation, which corresponds to an approximate horizontal resolution of 1.5km.*

*2. Per transect: for some parameters (e.g. particle filter measurements) only transect average observations exist. Other parameters (e.g. PSD) appeared to be constant over individual transects within the measurement uncertainty. Those were averaged to reduce measurement noise.*

*3. Fixed: for some parameters constant values are used for the entire flight, e.g. for those taken from literature or in the case of scarce data coverage."*

**I also wonder if some noise might be due to very small shifts in the time measurements of the individual instruments (i.e., time stamp on one instrument may not exactly match another instrument), which can be very important in plumes with strong horizontal variations. One could quickly check at the BB plume edge whether measurements line up in time.**

All instruments on NASA's DC8 were synchronized to the onboard time stamp. Consequently, even for the raw high temporal resolution (1s), we did not find systematic temporal shifts between the relevant instruments at plume edges. We therefore conclude that temporal shifts are a minor source of error for our 10s-resolution simulations.

**Line 392: It makes sense to use outside the plume observations for the background aerosol, but then the authors use a refractive index from the literature. Why not use a refractive index that may be more representative of the aerosol conditions outside of the plume?**

There were no direct measurements of aerosol refractive indices during FIREX-AQ. We used literature values for the background aerosol as they satisfied our accuracy requirements. Our sensitivity studies showed that variations in the real (imaginary) part of the background aerosol refractive index of 50 % (factor of 10) lead to less than 2% changes in the modeled actinic flux.

**Section 6: One remaining topic that could be discussed is whether there it is valuable to examine other FIREX flights. I am thinking of more complex situations in which clouds may be present. Do the authors think that examining one case is sufficient to evaluate the model?**

We find the Shady Fire case to be sufficient for a general validation of the model and the modeling approach. At the same time, we are aware that the Shady Fire is a particularly favorable case, and other cases (including scenarios with cloud cover) remain to be investigated in the future. To make this clearer we extended the corresponding discussion (see first answer to reviewer 2 below).

**Another topic that could be discussed is the implications for 3-D chemical transport models. Unless the active fire area is very large, BB plumes near their sources are not likely to be represented adequately by the coarse resolution of chemical transport models. The models will overly smooth these plumes, complicating how one evaluates computed photolysis rates and actinic fluxes with FIREX-AQ data.**

We addressed this during a response to a previous question above, by expanding the conclusions (L722) as follows:

*"Further applications are conceivable in the context of chemical transport modeling (CTM). To improve efficiency, CTMs typically run at relatively coarse spatial resolution (several km) and resort to simplifications (e.g. 1D-RT modeling) and parametrisations to account for photochemical processes. Detailed modeling based on accurate measurements of the atmospheric state, as presented in this study, can help to understand magnitudes and origins of the uncertainties introduced into CTMs by these approximations."*

**Review 2: comments and answers**

**The data origin is introduced by the authors. However, summary additional information about the 90 plumes in this database could be provided. In particular, the authors should explain the reasons for selecting the "Shady Fire" against the rest. They should indicate that this plume is representative enough of usual atmospheric conditions. Moreover, the authors could explain the atmospheric variables during this fire, such as wind speed and direction, temperature, synoptic pattern, ...**

We believe that the Shady Fire is a favorable case to validate the model. Similar investigations under less favorable conditions are still to be made. To make this clearer, we extended the corresponding paragraph accordingly::

*"Our case study focuses on measurements from the "Shady Fire" on July 25, 2019 in Idaho, which we chose for various reasons. Compared to other fires, the data coverage is high. The multiple-hour-long flight included three plume overflights as well as 20 plume transects during daylight and clear sky conditions. Most instruments*

*successfully collected data throughout the entire flight. The burned area over the sampling period was small ($\approx 2\ km^2$) and the burned fuel was homogeneous. The plume was large with an extent of about 10 km $\times$1 km $\times$ 100 km (W $\times$H $\times$L), ensuring good spatial sampling despite the high aircraft speed ($\approx$ 150 m/s) and justifying the 1D model assumption of a horizontally homogeneous atmosphere. Observed solar azimuth angles (SAAs) were between 250 $\circ$ and 290 $\circ$, and the Sun is therefore almost aligned with the plume axis (270 $\circ$ azimuthal orientation) over the entire flight (Fig. 2), which is a favorable configuration for avoiding horizontal radiative transport effects (Section 6). Wind conditions were very stable. At the sampling altitude (4200 to 5200 m MSL), transect-averaged wind speed and wind direction over the entire flight were (9 +/-3) m/s and (270 +/- 12) deg. At the same time, a constantly low relative humidity of (32+/-5) % prevented excessive hygroscopic growth of the aerosol particles. All in all, the Shady Fire represents a particularly favorable case. Even though it might not represent typical conditions, it is an ideal starting point for our purposes, as model validation and error analysis occur in a comparably controlled environment. Future applications of the presented modeling approach to other plumes under less favorable conditions are discussed in Section 7."*

We further extended the paragraph in lines 710-716.

*The Shady fire has been selected for its favorable conditions, such as the large plume size, clear sky conditions, comprehensive sampling, and the alignment of the sun with the plume axis. An expansion of our analysis to other FIREX-AQ plumes would allow assessment of actinic fluxes and RT modeling challenges in a wider variety of BB plumes. From the 90 fires sampled during FIREX-AQ, about five other fires provide similarly favorable conditions. Prevailing challenges for the study of other fires are scarce sampling and the presence of clouds.*

**Since the variables modelled are quite specific, the authors could increase the number of possible readers with the introduction of possible simple applications of this model. Moreover, the authors could indicate if this model could be used in the future by other researchers with a web-based application.**

A web-based version is not currently planned. We revised the first paragraph in the conclusions (L677) to point out the abilities of the model:

*"We have introduced and validated VPC, a VLIDORT-based quasi-spherical 1D RT model. VPC can calculate radiances, radiative fluxes and photolysis frequencies for a wide range of atmospheric conditions, including high loads of complex aerosol mixtures as they occur in BB or other plumes. VPC also efficiently calculates Jacobians of the simulated quantities with respect to the input parameters, facilitating its use as a forward model in remote sensing retrievals or similar inversion problems.*

*We have constrained the model by a comprehensive set of aerosol measurements performed during FIREX-AQ and calculated actinic fluxes and photolysis frequencies in BB plumes with an accuracy of 10-20% compared to direct measurements."*

**L. 247. A smoothing kernel is used. The authors could explain the used window in this kernel and its calculation procedure or the reason for such window.**

We revised the paragraph accordingly:

*"RT simulations can be performed for user-defined sets of wavelengths. Ideally, actinic flux spectra are calculated line-by-line, at a resolution resolving even narrow solar Fraunhofer and atmospheric absorption-lines (on the order of few pm in the UV-Vis). However, using very small wavelength intervals is inefficient. On the other hand, subsampling the wavelength range decreases the accuracy of the simulation (e.g. Madronich, 1990). A number of steps have therefore been taken to make VPC flexible and more efficient. The resolution of the originally highly-resolved ($\Delta\lambda \approx 0.01nm$) literature spectra used by the model (Section 2.2.3 and 2.2.4) can be reduced prior to simulation by Gaussian smoothing, i.e. convolution of the spectra with a Gaussian kernel of defined width, typically on the order of 1 nm FWHM (full width at half maximum). This option is useful for efficient calculation of outputs averaged over few nm wavelength intervals, at the cost of relatively small errors, introduced by the commutation of RT modeling and spectral smoothing. We found < 1% (< 3%) errors in the photolysis frequencies for a smoothing kernel of 1nm (2nm) FWHM and typical atmospheric scenarios."*

**Minor remarks**

**L. 118. Suppress one parenthesis.**

Suppressed.

**L. 935. Revise this reference.**

Revised.

**Supplement**

**L. 10. "to the" is repeated.**

Corrected.

**Figure S4. Introduce colour scale.**

Color scale added.

**Other modifications**

To improve the structure of the conclusion section, we moved the following paragraph from line 698 to line 680:

*"Previous model-measurement comparisons in clean atmospheres have found differences of < 10%. Considering the highly complex and inhomogeneous RT environment in dense BB plumes, the agreement between VPC and the observations is remarkably good."*

We added FWHM (full width at half maximum) to the table of abbreviations (Table A1).

We made minor spelling corrections to consistently use US-English.

In the conclusions, we changed tenses from simple past to past perfect, where applicable.

To improve the clarity for our motivation we changed the paragraph in L67 from:

*Continuous distributions of actinic flux and photolysis frequencies in BB plumes with the accuracy and spatial resolution needed for atmospheric chemistry studies can only be inferred using RT models (e.g. Decker, 2021; Palm, 2021).*

To:

*For atmospheric chemistry studies in BB plumes, a more complete picture is desirable, as the high spatial variability of actinic fluxes and photolysis frequencies significantly impacts plume processing (e.g., Decker, 2021; Palm, 2021). Continuous distributions of actinic flux and photolysis frequencies with the required accuracy and spatial resolution can only be inferred using RT models.*

L269: We removed the sentence "*In this study and in most other cases, this is only ozone.",* as it is out of place here and is repeated in the following paragraph.